# Predicting yield of individual field-grown rapeseed plants from rosette-stage leaf gene expression

Sam De Meyer[1,2‡], Daniel Felipe Cruz[1,2‡], Tom De Swaef[3], Peter Lootens[3], Jolien De Block[1,2], Kevin Bird[1,2¤], Heike Sprenger[1,2], Michael Van de Voorde[1,2], Stijn Hawinkel[1,2], Tom Van Hautegem[1,2], Dirk Inzé[1,2], Hilde Nelissen[1,2], Isabel Roldán-Ruiz[1,3], Steven Maere[1,2]*

1 Department of Plant Biotechnology and Bioinformatics, Ghent University, Ghent, Belgium, 2 VIB Center for Plant Systems Biology, Ghent, Belgium, 3 Plant Sciences Unit, Flanders Research Institute for Agriculture, Fisheries and Food (ILVO), Melle, Belgium

¤ Current address: Department of Plant Sciences, University of California-Davis, Davis, California, United States of America

‡ These authors share first authorship on this work.

* steven.maere@psb.vib-ugent.be

**Data Availability Statement:** The datasets and computer code produced in this study are available in the following databases: - RNA-seq data: S1 Data and ArrayExpress, E-MTAB-11904 (https://

## Abstract

In the plant sciences, results of laboratory studies often do not translate well to the field. To help close this lab-field gap, we developed a strategy for studying the wiring of plant traits directly in the field, based on molecular profiling and phenotyping of individual plants. Here, we use this single-plant omics strategy on winter-type *Brassica napus* (rapeseed). We investigate to what extent early and late phenotypes of field-grown rapeseed plants can be predicted from their autumnal leaf gene expression, and find that autumnal leaf gene expression not only has substantial predictive power for autumnal leaf phenotypes but also for final yield phenotypes in spring. Many of the top predictor genes are linked to developmental processes known to occur in autumn in winter-type *B. napus* accessions, such as the juvenile-to-adult and vegetative-to-reproductive phase transitions, indicating that the yield potential of winter-type *B. napus* is influenced by autumnal development. Our results show that single-plant omics can be used to identify genes and processes influencing crop yield in the field.

## Author summary

In the face of world population growth and climate change, the development of crops with increased yield and stress resilience is more urgent than ever. A major bottleneck in this process is translating the results of lab experiments to the field, in part because plant growth conditions in a lab are very different from field conditions. Here, we assess the merits of an alternative approach in which data is generated directly in the field, thereby bypassing the translation step. We profiled the gene expression and trait variability of a population of genetically (nearly) identical rapeseed plants grown in the same field, and

www.ebi.ac.uk/arrayexpress/experiments/E-MTAB-11904). - Phenotype data: S1 Data. - Weather station data: S1 Data. - Rosette and leaf images: Zenodo (https://zenodo.org/record/7072000). - Data analysis scripts: Zenodo (https://zenodo.org/record/7072000) and GitHub (https://github.com/MMichaelVdV/Brassica_segmentation).

**Funding:** SDM is a fellow of the Research Foundation-Flanders (FWO, grant 1146319N). The work of KB was supported by a Fulbright U.S. student award. The work of SH in the lab of SM was supported through a research collaboration with Inari Agriculture NV funded in part by Flanders Innovation & Entrepreneurship (VLAIO, grant HBC.2019.2814). The funders had no role in study design, data collection and analysis, decision to publish, or preparation of the manuscript.

**Competing interests:** The authors have declared that no competing interests exist.

used machine learning models to link the individual plants' gene expression to their phenotypic traits. We find that the plants' yield traits in spring can be predicted to a considerable extent from gene expression profiled > 5 months earlier in autumn. More importantly, we find that the top predictors in these models function in processes known to affect autumnal plant growth and development. This shows that our single-plant omics approach can be used to identify genes and processes influencing crop yield in the field.

## Introduction

One of the major aims of molecular biology research is to unravel how genes influence phenotypes. This usually involves applying perturbations to the genome or growth environment of an organism of interest and analyzing the ensuing molecular and phenotypic responses. Generally, well-chosen perturbations are applied in a controlled experimental setting, and technical and biological replicates are performed to allow for sufficiently powerful analyses despite noise in the data. Noise in this context may refer to measurement errors, noise due to uncontrolled factors in the experimental setup, or noise due to cellular or environmental stochasticity. The main purpose of avoiding or averaging out such noise is to facilitate causal interpretation of the link between a perturbation and its molecular and phenotypic effects.

It is becoming increasingly clear however that data noise caused by uncontrolled experimental factors and even purely stochastic effects can be a valuable source of information, instead of merely a nuisance. Several studies have shown that stochastic gene expression noise in single cells can be used to infer regulatory influences [1–3]. Gene networks are also increasingly inferred from single-cell gene expression datasets in which differences among cells are not purely due to stochastic effects in an otherwise homogeneous cell population, but reflect additional uncontrolled heterogeneity among cells, e.g. in the temporal progression of a cell differentiation program [4–10].

In addition, several studies have investigated the information content of 'noise' datasets in which the profiled entities are multicellular individuals rather than single cells. Bhosale *et al.* [11] found that gene expression noise among individual *Arabidopsis thaliana* plants grown under the same conditions harbored as much information on the function of genes as gene expression responses to controlled perturbations. The dataset analyzed by Bhosale *et al.* [11] was however not ideal because it contained data on plants of three different accessions grown in six different labs [12], causing lab and accession effects that had to be removed computationally to uncover the individual plant noise of interest. Recently, a study on a cleaner *A. thaliana* seedling dataset confirmed that gene expression noise among individuals of the same background grown under the same lab conditions contains useful information on gene functions and regulatory relationships [13].

A common denominator in the aforementioned studies is that even under controlled conditions, each cell or individual is subject to a set of stochastic or other perturbations that escape experimental control, and that these uncontrolled perturbations, like any perturbations, generate responses that contain valuable information on the wiring of gene networks. Although most studies to date focused on the information content of noise under controlled lab conditions, there is no reason to believe that 'noise' datasets generated under less controlled conditions would be less valuable. On the contrary, studies performed in a more natural setting in which organisms are subject to uncontrolled perturbations may yield information that cannot easily be recovered from experiments under controlled lab conditions.

In the plant sciences for instance, controlled growth conditions in a laboratory are generally very different from field conditions, in which plants are subject to a plethora of highly variable environmental cues that often have non-additive phenotypic effects [14–21]. Results obtained in the laboratory therefore often translate poorly to the field [14, 22–26]. Narrowing this lab-field gap is essential to speed up the development of new crop varieties and optimized agricultural practices, both of which are direly needed in view of the current challenges posed by world population growth, land use and climate change. One option to narrow the lab-field gap is to make lab conditions more field-like [22], but the decreased experimental control this implies challenges traditional experimental design practices to e.g. ensure reproducibility. Another option is to perform interventional experiments in the field rather than the lab, but controlled interventions in a field may be costly and the level of control that can be achieved is often limited [22]. Observational 'uncontrolled perturbation' studies on the other hand can easily be set up in the field. Observational data come with their own array of challenges however, e.g. that many of the perturbations influencing the study subjects may remain unobserved and hence unknown, and that it is generally much more challenging to establish cause-effect relationships from observational data [27]. Nevertheless, even purely correlational data generated in the field may help narrow the lab-field gap in plant sciences.

To assess the information content of plant molecular responses to uncontrolled perturbations occurring in a field environment, we previously generated transcriptome and metabolome data on the primary ear leaf of 60 individual *Zea mays* (maize) plants of the same genetic background grown in the same field [28]. Similar to what was found for lab-grown *A. thaliana* plants [11], the transcriptomes of the individual field-grown maize plants were found to contain as much information on maize gene function as transcriptomes profiling the response of maize plants to controlled perturbations in the lab. In addition, we found that the single-plant transcriptome and metabolome data had better-than-random predictive power for several phenotypes that were measured for the individual plants, and the prediction models also produced sensible candidate genes for these phenotypes [28]. However, only a few phenotypes were measured in this study, and they were either closely associated with the material sampled for molecular profiling, not fully developed or both.

Here, we investigate in more detail how much phenotype information can be extracted from the transcriptomes of single plants subject to uncontrolled perturbations under field conditions. To this end, we profiled the rosette-stage leaf transcriptome of individual field-grown plants of the winter-type accession Darmor of *Brassica napus* ssp. *napus* (rapeseed), an important oilseed crop [29]. Additionally, a wide range of phenotypes was measured for all plants throughout the growing season. We find that the autumnal leaf transcriptomes of the individual plants do not only have predictive power for autumnal leaf phenotypes but also for yield phenotypes measured more than 5 months later, such as silique count and total seed weight. Furthermore, we find that many of the genes that feature prominently in our predictive models are related to developmental processes known to occur in autumn in winter-type *Brassica napus*, in particular the juvenile-to-adult and vegetative-to-reproductive phase transitions. Our results suggest that micro-environmental variations across the field cause a gradual buildup of developmental differences among plants that ultimately result in yield differences at the end of the growing season.

## Results

### Field trial, expression profiling and phenotyping

One hundred *Brassica napus* plants of the winter-type accession Darmor were grown in a field in a 10x10 equispaced grid pattern with 0.5 m distance between rows and columns (**Fig 1**). On

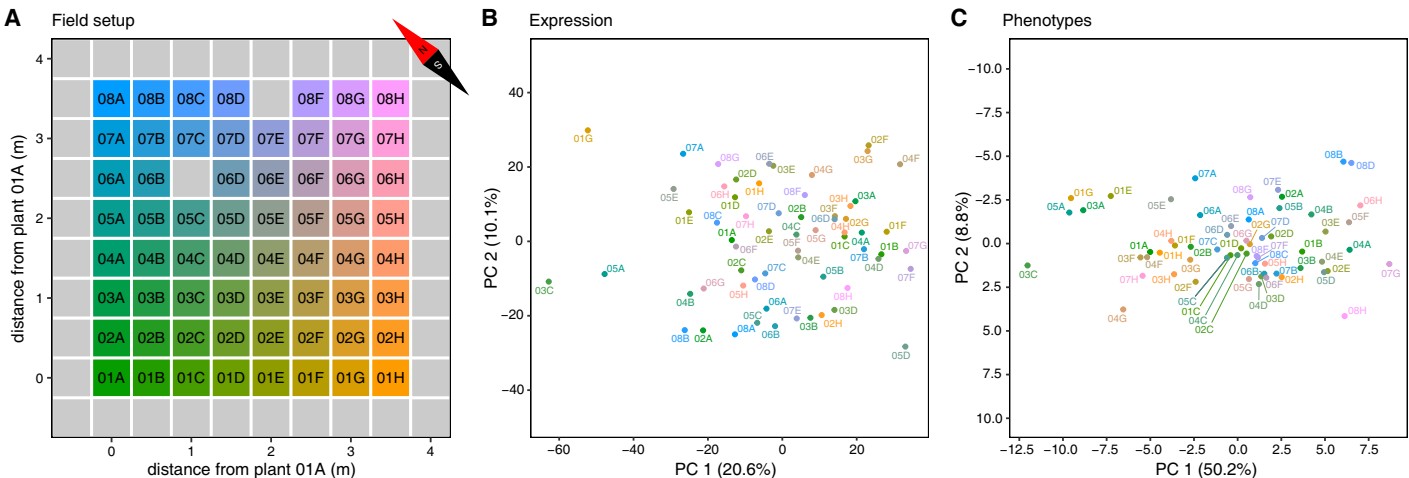

**Fig 1. Field trial layout and PCA plots for gene expression and phenotypes. A.** Plants were sown on a 10x10 equispaced grid with 0.5 m between rows and columns. Plant identifiers combine a number indicating the row (01–08) and a letter indicating the column (A-H) in which the plant was sown. Only plants with leaf 8 gene expression and phenotype profiles are labeled, border plants and grid positions at which no plants emerged are indicated by grey squares. **B.** Plot of the first two principal components of the leaf 8 gene expression dataset, after normalization and RNA-seq batch correction (see Methods). **C.** Plot of the first two principal components in the phenotype dataset. Individual plants in **B** and **C** are colored according to the color gradient in **A**, with similar coloring of plants indicating spatial proximity in the field.

November 28, 2016, the eighth rosette leaf (leaf 8) of 62 non-border plants was harvested, and the harvested leaves were expression-profiled individually (see Methods and **S1 Data**). After leaf sampling, the plants were allowed to overwinter and set seed in spring. 62 phenotypes were recorded for all plants, ranging from rosette areas and individual leaf measurements in autumn to root and shoot measurements at harvest the following spring (**S1 Data**). Likely because of the low planting density, many of the plants developed one or more secondary inflorescence stems at ground level, which is not usually observed for *B. napus* grown under lab conditions or in the field at agronomically relevant planting densities. These secondary stems (further referred to as side stems) were harvested separately from the primary inflorescence stem with its cauline secondary inflorescences (further referred to as stem 1). Several yield phenotypes were measured for both stem 1 and the entire shoot (i.e. stem 1 plus side stems), including dry weight, seed weight, seed count and silique count. Cauline secondary inflorescence stems on stem 1 and tertiary inflorescence stems on the side stems (both further referred to as branches) were also counted, and branch counts are reported for both stem 1 and the entire shoot (the latter being the sum of branch counts on stem 1 and the side stems). Shoot growth phenotypes such as the time of maximum shoot growth, the maximum shoot growth rate and the end of shoot growth were derived from plant height time series data through curve fitting (see Methods). Several phenotypes were defined as ratios of other phenotypes, e.g. the ratio of total seed weight to shoot dry weight and the ratio of the total number of seeds to the total number of siliques per plant.

## Exploratory data analysis

Principal component analysis (PCA) suggests that there are no clearly separated subpopulations of plants with distinct expression or phenotype profiles (**Fig 1**). A few relative outliers are visible however, e.g. plants 03C and 04G in the phenotype PCA plot (**Fig 1C**), both small plants that yielded few seeds (**S1 Data**). Single-nucleotide polymorphism (SNP) analysis of the RNA-seq data (see Methods) did not uncover signs of substantial genetic substructure in the plant population (**S1 Fig**).

 

On the other hand, mapping of the field coordinates on the expression and phenotype PCA plots (**Fig 1**) and analysis of the correlation of the distance between plants in transcriptome or phenotype space with the physical distance between plants in the field (**S2 Fig**) suggest that there is spatial structure in the data. However, the levels of only 140/76,808 transcripts (0.18%) and 1/41 phenotypes (2.44%, root system width) were found to be significantly spatially autocorrelated across the field (Moran's I, Benjamini-Hochberg (BH) adjusted permutation test $p$-values ($q$-values) $\leq$ 0.05, **S2 Data** and **S3 Fig**). In a previous study on a similar number of field-grown maize plants [28], 14.17% of transcripts were found to be significantly spatially autocorrelated at $q \leq 0.01$, which is considerably more than the 0.18% recovered here at $q \leq 0.05$. This is mostly due to differences in the way Moran's I values and their significance were calculated in the present study versus Cruz *et al.* [28] (use of a queen contiguity-based spatial weight matrix and permutation-based testing for determining $p$-values in the present study, versus an inverse distance-based spatial weight matrix and parametric testing in Cruz *et al.* [28], see Methods and **S2 Data**) and to a stronger effect of BH $p$-value adjustment for *B. napus* than for maize (76,808 transcripts were tested for *B. napus* versus 18,171 for maize). Using the method from Cruz *et al.* [28] on the present dataset yields 5,371/76,808 (6.99%) significantly spatially autocorrelated transcripts and 2/41 (4.88%) significantly spatially autocorrelated phenotypes at $q \leq 0.05$ (**S2 Data**). The Spearman rank correlations between the Moran's I values obtained by both methods are 0.8761 and 0.8645 for gene expression profiles and phenotype profiles, respectively. Although the method from Cruz *et al.* [28] yields more significant spatial autocorrelations, we consider the method using non-parametric permutation testing to be more reliable, as the parametric method assumes a normal distribution of the Moran's I value under the assumption of no spatial autocorrelation, which does not always hold. The results presented below are therefore based on the method using a queen contiguity-based spatial weight matrix and permutation-based testing.

To assess whether some functional classes of genes have on average a stronger or weaker spatial autocorrelation signal than other classes, regardless of the statistical significance of the Moran's I values, two-sided Mann-Whitney U (MWU) tests [30] were performed on the transcript list ranked in order of decreasing Moran's I value. Genes involved in e.g. photosynthesis, translation, the response to abiotic stimuli, response to cytokinin, regulation of circadian rhythm, photoperiodism and the vegetative to reproductive phase transition were found to have a significantly higher Moran's I on average than other genes (MWU $q \leq 0.05$, **S2 Data**). This suggests that there is spatial patterning in the data, but that its discovery may be hampered by a lack of statistical power due to the small size of the field trial.

Most continuous phenotypes and high-count discrete phenotypes (e.g. seed and silique counts) are at least approximately normally distributed (Anderson-Darling and Shapiro-Wilk normality tests, $p > 0.01$, **S3 Data**), with the exception of five ratio phenotypes (seeds per silique, seeds per silique stem 1, seed weight/dry weight stem 1, total seed weight/shoot dry weight and branches per stem), leaf count (74 DAS) and two shoot growth phenotypes (time of max shoot growth and end of shoot growth). Many of these phenotypes exhibit relative outliers that may influence normality testing results (**S4 Fig**). When removing outliers (see Methods), four additional phenotypes (seeds per silique, seeds per silique stem 1, stem 1 seed weight/stem 1 dry weight, total seed weight/shoot dry weight) were found to be approximately normally distributed (Anderson-Darling and Shapiro-Wilk normality test, $p > 0.01$, **S3 Data**).

Some phenotypes were found to be more variable across the field than others. Dry weight, seed and silique phenotypes at harvest are the most variable, with coefficients of variation (CVs) between 43.7% and 51.9% (**S3 Data**). Taproot length also has a high CV (42.8%). Plant height (278 DAS) and shoot growth parameters exhibit the lowest CV values ($< 7\%$). Most ratio phenotypes also have relatively low CV values ($\leq 20.3\%$), with the exception of siliques per branch (35.3%), siliques per branch stem 1 (35.0%) and branches per stem (33.6%). When

 

removing outliers, the CV of some of these ratio phenotypes is further reduced, notably for seed weight stem 1/dry weight stem 1 (20.3% → 9.5%), total seed weight/shoot dry weight (18.1% → 8.9%), seeds per silique (19.5% → 14.7%) and seeds per silique stem 1 (19.4% → 14.6%). Leaf and branch phenotypes generally exhibit intermediate CVs. Whereas leaf 8 fresh weight (81 DAS), leaf 8 area (81 DAS), total branch count and rosette area (42 DAS) have a CV $\geq$ 30%, other leaf 8 and leaf 6 phenotypes and branch count stem 1 exhibit a CV in the range 19.1%-23.2%, and leaf 8 chlorophyll content (81 DAS) has a CV of only 12.5%.

Gene expression also exhibits substantial variability across the field. Ignoring genes expressed in less than 10 samples, the median gene has an expression CV of 34.2% (**S3 Data**). To investigate whether some classes of genes vary more in expression than others across the field, we ranked *B. napus* genes based on a normalized version of their expression CV (*normCV*, see Methods and **S3 Data**). MWU tests [30] were performed to assess whether any Gene Ontology (GO) biological processes are represented more at the top or bottom of the *normCV*-ranked gene list than expected by chance (**S3 Data**). As observed in earlier studies on populations of lab-grown *Arabidopsis thaliana* Col-0 plants [31] and field-grown *Zea mays* B104 plants [28], genes involved in photosynthesis and responses to biotic and abiotic stimuli were found to be on average more variably expressed than other genes, while genes involved in housekeeping functions related to protein, RNA and DNA metabolism were found to be on average more stably expressed across the field (**S3 Data**). To what extent high gene expression variability is due to either variability in the levels of external stimuli experienced by the individual plants or due to a higher intrinsic noisiness of a gene's expression levels (on the scale of entire leaves) is unclear. Some categories of genes with more variable expression across the field, such as genes involved in photosynthesis or response to abiotic stimuli, also exhibit higher Moran's I values on average, suggesting that their variability may be linked to external stimuli that are spatially patterned. On the other hand, most genes with highly variable expression do not exhibit strong spatial patterns (**S5 Fig**), which indicates that their expression variability may be caused by intrinsic stochastic factors, or alternatively by extrinsic factors that are not spatially autocorrelated at the field sampling resolution employed.

## Linking phenotypes to the leaf 8 expression profiles of single genes

In order to get an overall view on which leaf 8 gene expression profiles significantly relate to which phenotypes, linear mixed-effects (lme) models were used to assess the relation between each gene expression profile and each phenotype separately (further referred to as single-gene models). lme models were used for this instead of ordinary linear models because they can take into account spatial autocorrelation effects (see Methods), which can bias significance testing in ordinary linear models [32]. Between 11,986 and 14,032 gene expression profiles, out of 76,808, were found to be significantly associated ($q \leq 0.05$) with leaf 8 phenotypes such as leaf 8 length, width, area and fresh weight (**Table 1** and **S4 Data**). That leaf 8 phenotypes yield more associated genes than other phenotypes is not surprising, given that leaf 8 was used for gene expression profiling. Next to leaf 8 phenotypes, also other leaf and rosette phenotypes feature more associated genes than non-leaf phenotypes, except for leaf 6 length (74 DAS). The gene sets associated with leaf phenotypes are generally significantly enriched (hypergeometric test, $q \leq 0.05$) in genes involved in e.g. response to biotic and abiotic stimuli (salt), photosynthesis, circumnutation, cell wall biogenesis, amino acid metabolism and response to sulfate and nitrogen starvation (**S5 Data**). Additionally, leaf phenotype-related gene lists show significant enrichment, notably among transcription factors, in genes involved in dorsal/ventral, adaxial/abaxial and radial pattern formation, phloem, xylem and procambium histogenesis, and meristem development (**S5 Data**).

**Table 1. Numbers of significant gene expression-phenotype associations.**

| Phenotype | All genes | | | Transcription factors | | |
|---|---|---|---|---|---|---|
| | # Significant | Most significant | q | # Significant | Most significant | q |
| leaf 8 length (76 DAS) | 14032 | BnaC07g39340D | 4.35E-14 | 453 | BnaA05g33840D | 4.44E-09 |
| leaf 8 width (76 DAS) | 13695 | BnaC07g39340D | 1.62E-15 | 429 | BnaA05g33840D | 6.98E-09 |
| leaf 8 width (81 DAS) | 13605 | BnaA02g18860D | 1.37E-12 | 420 | BnaA05g33840D | 1.36E-08 |
| leaf 8 fresh weight (81 DAS) | 12989 | BnaC07g39340D | 5.79E-12 | 412 | BnaAnng02740D | 4.17E-08 |
| leaf 8 area (81 DAS) | 12569 | BnaC07g39340D | 1.29E-13 | 408 | BnaAnng02740D | 1.64E-08 |
| leaf 8 length (81 DAS) | 11986 | BnaA01g14450D | 6.39E-14 | 383 | BnaA05g27750D | 5.25E-08 |
| leaf count (74 DAS) | 10442 | BnaC04g49060D | 1.86E-07 | 313 | BnaA05g33840D | 2.16E-06 |
| rosette area (42 DAS) | 7196 | BnaC09g39140D | 7.14E-06 | 212 | BnaA06g39930D | 1.06E-04 |
| leaf 6 width (74 DAS) | 5386 | BnaA09g04980D | 2.05E-05 | 184 | BnaA05g27750D | 2.76E-04 |
| time of max shoot growth | 3498 | BnaC06g28860D | 7.29E-07 | 89 | BnaC04g03950D | 5.42E-06 |
| total shoot dry weight | 1859 | BnaA05g29010D | 1.37E-06 | 76 | BnaCnng05590D | 2.36E-04 |
| total shoot dry weight (w/o seeds) | 1802 | BnaA05g29010D | 8.88E-07 | 68 | BnaAnng37500D | 4.89E-04 |
| dry weight stem 1 | 1612 | BnaA05g29010D | 2.79E-06 | 72 | BnaC01g37260D | 5.87E-05 |
| dry weight stem 1 (w/o seeds) | 1611 | BnaA05g29010D | 6.82E-06 | 75 | BnaA08g12050D | 3.13E-04 |
| total seed weight | 1598 | BnaA06g35450D | 1.02E-05 | 66 | BnaCnng05590D | 6.05E-05 |
| total seed count | 1545 | BnaA06g35450D | 9.10E-06 | 63 | BnaCnng05590D | 1.12E-04 |
| seed weight stem 1 | 1539 | BnaA05g29010D | 1.65E-05 | 66 | BnaC01g37260D | 1.65E-05 |
| total silique count | 1520 | BnaA06g35450D | 3.92E-05 | 64 | BnaCnng05590D | 7.12E-04 |
| seed count stem 1 | 1449 | BnaC01g37260D | 2.58E-05 | 67 | BnaC01g37260D | 2.58E-05 |
| leaf 6 length (74 DAS) | 1345 | BnaA09g04980D | 2.37E-04 | 34 | BnaA02g18720D | 8.39E-03 |
| silique count stem 1 | 1248 | BnaA05g29010D | 6.32E-05 | 56 | BnaC01g37260D | 6.32E-05 |
| branch count stem 1 | 1110 | BnaA06g35450D | 2.52E-04 | 39 | BnaC01g37260D | 1.95E-03 |
| siliques per branch stem 1 | 593 | BnaA01g17100D | 2.24E-03 | 29 | BnaC01g37260D | 2.48E-03 |
| total seed weight/shoot dry weight | 458 | BnaC09g50070D | 2.24E-05 | 13 | BnaC04g55440D | 4.81E-04 |
| seed weight stem 1/dry weight stem 1 | 280 | BnaA03g50380D | 8.90E-04 | 6 | BnaC03g62970D | 3.39E-03 |
| branch count stem 1/length stem 1 | 240 | BnaAnng11300D | 8.22E-03 | 5 | BnaC01g37260D | 4.04E-02 |
| seeds per silique stem 1 | 233 | BnaC05g45470D | 8.73E-04 | 9 | BnaC04g03950D | 4.42E-03 |
| seeds per silique | 112 | BnaA08g07570D | 9.01E-04 | 3 | BnaC04g03950D | 1.43E-02 |
| plant height (278 DAS) | 99 | BnaA06g34140D | 7.20E-04 | 5 | BnaC01g37260D | 2.48E-02 |
| total branch count | 89 | BnaA06g35450D | 2.39E-03 | 1 | BnaCnng05590D | 5.82E-03 |
| root system width | 4 | BnaA01g06800D | 7.53E-03 | 0 | - | - |
| siliques per branch | 3 | BnaAnng39720D | 2.22E-02 | 0 | - | - |
| branches per stem | 0 | - | - | 0 | - | - |
| taproot length | 0 | - | - | 0 | - | - |
| leaf 8 chlorophyll content (81 DAS) | 0 | - | - | 0 | - | - |
| max shoot growth rate | 0 | - | - | 0 | - | - |
| end of shoot growth | 0 | - | - | 0 | - | - |
| rosette lesions (74 DAS) | 0 | - | - | 0 | - | - |
| leaf 6 lesions (74 DAS) | 0 | - | - | 0 | - | - |
| leaf 8 lesions (76 DAS) | 0 | - | - | 0 | - | - |
| stem count | 0 | - | - | 0 | - | - |

Table legend: For any given phenotype, results are reported on the complete gene set ($n$ = 76,808; 'All genes' columns) and on the set of transcription factors ($n$ = 2,521; 'Transcription factors' columns). In both cases, the results shown include (from left to right) the total number of significant gene expression-phenotype associations ($q \leq 0.05$), the most significant gene and its $q$-value.

Interestingly, appreciable numbers of gene-phenotype associations were found as well for several phenotypes that are only distantly related in space and time to the leaf 8 material profiled for RNA-seq. In particular seed, silique and shoot dry weight phenotypes yielded high numbers of associated genes, ranging from 1,859 genes for total shoot dry weight to 1,248 genes for the silique count on stem 1 at harvest (**Table 1** and **S4 Data**). Many of the gene sets associated with these phenotypes are enriched in genes involved in nitrate assimilation, superoxide metabolism, circumnutation, circadian rhythm, response to biotic and abiotic stimuli (cold, salt, water deprivation), response to nutrient levels (nitrogen, sulphate and phosphate starvation), and, in particular among transcription factors, phosphate ion homeostasis, histone modification, regulation of the vegetative to reproductive phase transition and floral organ morphogenesis (**S5 Data**). 1,110 genes were found associated with the branch count on stem 1, with GO enrichments similar to those obtained for dry weight, silique and seed phenotypes (**S5 Data**). In contrast, the total branch count phenotype only yields a set of 89 associated genes (**Table 1** and **S4 Data**), which is however also strongly enriched in e.g. superoxide metabolism and salt stress genes. The fact that the total branch count is composed of cauline secondary inflorescence stems on stem 1 and tertiary inflorescence stems on the side stems may render this phenotype less relevant. No genes were found associated ($q \leq 0.05$) with taproot length and only four with root system width, suggesting that autumnal leaf 8 gene expression may not contain a lot of information on root phenotypes. On the other hand, given the difficulty of recovering intact root systems from the soil, it is not excluded that root measurement errors may have influenced these results.

Phenotypes with very low CV such as leaf 8 chlorophyll content (81 DAS), the maximum shoot growth rate and end of shoot growth yielded no significantly associated genes, suggesting that the biological variation of these phenotypes is limited and that the observed variation may be dominated by technical noise (**Table 1** and **S4 Data**). The phenotype with the lowest CV on the other hand, the time of maximal shoot growth (CV = 0.6%), features 3,498 significant leaf 8 gene expression correlates. The associated gene set is strongly enriched in genes involved in e.g. cell wall biogenesis and response to biotic stimuli (**S5 Data**). For plant height (278 DAS) (CV = 6.8%), 99 associated genes are found with minor GO enrichments.

Ratio phenotypes exhibit between 0 and 593 associated genes. In particular the branches per stem and siliques per branch ratios do poorly (0 and 3 associated genes, respectively). Both involve the total branch count, which is itself only associated with 89 genes. Ratios involving the branch count on stem 1 on the other hand yield between 240 and 593 associated genes. One potential reason for ratio phenotypes having at most a few hundred gene associations is that ratios suffer from increased error levels due to the propagation of measurement errors from both the numerator and denominator. This may be particularly problematic for ratios of highly correlated variables such as the seeds per silique and seed weight/dry weight phenotypes (both for stem 1 and the entire shoot), which exhibit a low CV and likely have even lower true biological variation. No genes were found associated at $q \leq 0.05$ with qualitative or low-count discrete phenotypes such as rosette lesions (74 DAS), leaf 6 lesions (74 DAS), leaf 8 lesions (76 DAS) and stem count (i.e. stem 1 plus the number of side stems).

### Leaf and final yield phenotypes of individual field-grown *B. napus* plants can be predicted to a considerable extent from their autumnal leaf 8 transcriptome

Most phenotypes feature many significantly related gene expression profiles (**Table 1**). To assess to what extent phenotypes can be predicted from the entire leaf 8 transcriptome, we used models using all gene expression profiles as predictive features, or a sizeable subset

thereof (further referred to as multi-gene models). As the feature space (76,808 gene expression profiles) is huge compared to the number of samples (62 plants), simply using ordinary linear models or lme models with all gene expression values as features makes no sense, as there's a myriad different ways to perfectly explain a given phenotype for 62 plants from 76,808 gene expression profiles. Most of these models will be seriously overfit and will generalize badly to unseen data. Instead, we used elastic net [33] and random forest [34] models. Elastic net (enet) is a regularized regression method that uses a penalty on the sum of feature coefficient magnitudes (L1-norm, as in lasso) and an additional penalty on the magnitude of the coefficient vector (L2-norm or Euclidean norm, as in ridge regression) to shrink the feature coefficients and limit the amount of features used to explain an outcome variable, with the purpose of reducing overfitting and hence enhancing the generalizability of the resulting model to unseen data. Random forest (RF) reduces overfitting by training a 'forest' of decision trees, in which each tree is trained on a dataset sampled with replacement from the original data, and at each tree split only a random subset of features are taken into account as possible predictors. The predicted outcome for a given input is then taken to be the average of the output of the different trees in the forest. The main reason for using both enet and RF models is that the former can only capture linear relationships in the data, while the latter can also capture non-linear effects.

Models were learned using either the set of all genes (n = 76,808) or the (much smaller) set of all transcription factors (TFs, n = 2,521) as features. In both cases, to further reduce the number of features used to train the machine learning models, we used a priori feature selection to select subsets of features that are potentially interesting. As different feature selection techniques may have different biases, we used three different feature selection techniques, namely HSIC lasso [35], a Spearman correlation filter and a filter selecting only genes with median *rlog* gene expression $> 0$ (see Methods).

To assess to what extent the models generalize to unseen data, the enet and RF models were trained in a 10-fold cross-validation setup (i.e. the data was split in 10 data subsets, each set was selected in turn as the test set, the model was trained on the remaining 9 sets and applied to the test set, see Methods). To assess the variance of the test set predictions, the 10-fold cross-validation procedure was repeated 9 times with different splits, giving rise to 90 test sets and 9 test set predictions per plant for each combination of phenotype, model type (RF or enet), potential feature set (all genes or TFs) and feature selection technique. The best model for a given phenotype and potential feature set was taken to be the one with the highest median test $R^2$ value across all 90 test sets for continuous and high-count phenotypes (see Methods), or the highest median test accuracy for qualitative or low-count discrete phenotypes (**Table 2** and **S6 Data**).

Not surprisingly, leaf 8 phenotypes, which are most closely related in space and time to the material sampled for transcriptome profiling, are the most predictable. Except for the leaf 8 chlorophyll content at sampling time (81 DAS), which features very poor prediction performance, the median test $R^2$ scores for leaf 8 phenotypes range from 0.48 to 0.70 when using all genes as potential features. Other leaf-related phenotypes such as leaf 6 width (74 DAS, median test $R^2$ = 0.38), rosette area (42 DAS, median test $R^2$ = 0.23) and leaf 6 length (74 DAS, median test $R^2$ = 0.07) are comparatively less predictable.

Surprisingly, many of the final seed, silique and shoot dry weight phenotypes are more predictable from the autumnal leaf 8 transcriptome than leaf 6 and rosette phenotypes, with seed weight on stem 1 rivaling the leaf 8 phenotypes in terms of median test $R^2$ value (**Table 2, S6 Data** and **Figs 2** and **S6**). All seed weight, seed and silique count and shoot dry weight phenotypes have median test $R^2$ values in the range [0.35, 0.51] for the all-genes models, which is in all cases higher than the 95th percentile of test $R^2$ values obtained from single train-test splits

**Table 2. Best-performing multi-gene and single-gene models for each phenotype.**

| Continuous and high-count phenotypes | All genes | | | | Transcription factors | | | | Single gene | | | |
|---|---|---|---|---|---|---|---|---|---|---|---|---|
| | Feature sel. | Model type | Median test R2 | Median pooled PCC | Feature sel. | Model type | Median test R2 | Median pooled PCC | Top gene | Median test R2 | Median pooled PCC | CV |
| leaf 8 width (76 DAS) | median | enet | 0.70 * | 0.87 | median | enet | 0.64 * | 0.84 | BnaC04g39580D | 0.67 | 0.83 | 2.32E-01 |
| leaf 8 width (81 DAS) | median | enet | 0.65 * | 0.86 | median | enet | 0.65 * | 0.84 | BnaA02g18860D | 0.62 | 0.83 | 2.32E-01 |
| leaf 8 area (81 DAS) | median | enet | 0.63 * | 0.83 | median | enet | 0.58 * | 0.81 | BnaCnng33420D | 0.60 | 0.81 | 3.70E-01 |
| leaf 8 fresh weight (81 DAS) | median | enet | 0.59 * | 0.81 | median | enet | 0.53 * | 0.78 | BnaCnng33420D | 0.58 | 0.79 | 3.88E-01 |
| seed weight stem 1 | spearman | enet | 0.51 * | 0.77 | median | enet | 0.53 * | 0.78 | BnaA05g29010D | 0.42 | 0.67 | 4.51E-01 |
| leaf 8 length (76 DAS) | spearman | rf | 0.51 * | 0.80 | median | enet | 0.53 * | 0.79 | BnaC07g39340D | 0.58 | 0.80 | 2.21E-01 |
| leaf 8 length (81 DAS) | spearman | rf | 0.48 * | 0.78 | spearman | enet | 0.51 * | 0.78 | BnaC01g17020D | 0.52 | 0.81 | 2.11E-01 |
| seed count stem 1 | spearman | enet | 0.47 * | 0.72 | median | enet | 0.43 * | 0.73 | BnaA06g20870D | 0.38 | 0.61 | 4.71E-01 |
| silique count stem 1 | median | enet | 0.46 * | 0.74 | median | enet | 0.45 * | 0.72 | BnaA05g29010D | 0.37 | 0.66 | 4.37E-01 |
| total seed count | spearman | enet | 0.45 * | 0.73 | median | enet | 0.38 * | 0.71 | BnaC03g60710D | 0.39 | 0.61 | 4.78E-01 |
| dry weight stem 1 | spearman | enet | 0.44 * | 0.73 | median | enet | 0.39 * | 0.70 | BnaA05g29010D | 0.40 | 0.70 | 4.83E-01 |
| dry weight stem 1 (w/o seeds) | hsic-5000 | enet | 0.42 * | 0.69 | spearman | enet | 0.35 * | 0.64 | BnaA05g29010D | 0.39 | 0.69 | 5.19E-01 |
| total seed weight | spearman | enet | 0.42 * | 0.74 | median | enet | 0.40 * | 0.70 | BnaA06g35450D | 0.41 | 0.69 | 4.69E-01 |
| total shoot dry weight | spearman | enet | 0.41 * | 0.71 | median | enet | 0.31 * | 0.68 | BnaA09g48720D | 0.41 | 0.67 | 4.89E-01 |
| leaf 6 width (74 DAS) | median | enet | 0.38 * | 0.68 | hsic-5000 | rf | 0.07 | 0.51 | BnaC03g15540D | 0.35 | 0.61 | 1.91E-01 |
| total silique count | median | enet | 0.38 * | 0.68 | median | enet | 0.36 * | 0.68 | BnaC04g21390D | 0.40 | 0.63 | 4.56E-01 |
| siliques per branch stem 1 | hsic-5000 | enet | 0.37 * | 0.67 | hsic-5000 | rf | 0.14 | 0.51 | BnaC04g21390D | 0.25 | 0.60 | 3.50E-01 |
| total shoot dry weight (w/o seeds) | spearman | enet | 0.35 * | 0.66 | spearman | enet | 0.29 | 0.62 | BnaA06g05150D | 0.40 | 0.69 | 5.13E-01 |
| leaf count (74 DAS) | median | rf | 0.24 * | 0.66 | median | rf | 0.40 * | 0.72 | BnaA01g34700D | 0.38 | 0.70 | 1.12E-01 |
| rosette area (42 DAS) | median | enet | 0.23 * | 0.59 | median | enet | 0.36 * | 0.68 | BnaC05g30740D | 0.33 | 0.64 | 3.00E-01 |
| branch count stem 1 | spearman | rf | 0.19 | 0.56 | median | enet | 0.11 * | 0.52 | BnaA10g29560D | 0.38 | 0.56 | 2.00E-01 |
| siliques per branch | spearman | enet | 0.16 | 0.49 | spearman | enet | -0.01 | 0.36 | BnaA08g09860D | 0.12 | 0.53 | 3.53E-01 |
| plant height (278 DAS) | median | enet | 0.12 * | 0.47 | spearman | enet | 0.16 | 0.51 | BnaC07g25920D | 0.34 | 0.63 | 6.81E-02 |
| total branch count | median | rf | 0.10 | 0.40 | median | enet | 0.17 * | 0.57 | BnaA09g48720D | 0.26 | 0.59 | 3.42E-01 |

(*Continued*)

**Table 2.** (*Continued*)

| Continuous and high-count phenotypes | All genes | | | | Transcription factors | | | | Single gene | | | |
|---|---|---|---|---|---|---|---|---|---|---|---|---|
| | Feature sel. | Model type | Median test R2 | Median pooled PCC | Feature sel. | Model type | Median test R2 | Median pooled PCC | Top gene | Median test R2 | Median pooled PCC | CV |
| branch count stem 1/length stem 1 | median | rf | 0.07 | 0.39 | median | rf | -0.06 | 0.23 | BnaC05g15590D | 0.22 | 0.55 | 1.58E-01 |
| leaf 6 length (74 DAS) | median | enet | 0.07 * | 0.45 | median | rf | 0.06 | 0.43 | BnaA09g04980D | 0.31 | 0.64 | 1.96E-01 |
| max shoot growth rate | median | rf | 0.03 | 0.41 | hsic-5000 | rf | -0.02 | 0.34 | BnaA10g21770D | 0.15 | 0.50 | 6.75E-02 |
| root system width | median | rf | 0.01 | 0.36 | median | rf | 0.17 * | 0.56 | BnaC07g01150D | 0.17 | 0.56 | 2.16E-01 |
| time of max shoot growth | median | rf | -0.02 | 0.38 | hsic-5000 | rf | 0.20 | 0.57 | BnaA05g08250D | 0.15 | 0.56 | 6.45E-03 |
| taproot length | spearman | rf | -0.02 | 0.26 | median | enet | -0.09 | 0.18 | BnaA04g17830D | 0.16 | 0.50 | 4.28E-01 |
| branches per stem | median | rf | -0.09 | 0.26 | hsic-5000 | rf | -0.12 | 0.21 | BnaC03g42190D | 0.13 | 0.51 | 3.36E-01 |
| leaf 8 chlorophyll content (81 DAS) | median | enet | -0.14 | -0.35 | median | enet | -0.12 | -0.40 | BnaA03g40350D | 0.09 | 0.46 | 1.25E-01 |
| seeds per silique | median | rf | -0.14 | -0.18 | median | enet | -0.17 | -0.31 | BnaC03g38990D | -0.09 | 0.22 | 1.95E-01 |
| seeds per silique stem 1 | median | enet | -0.15 | -0.31 | median | enet | -0.15 | -0.08 | BnaC01g44890D | -0.05 | 0.40 | 1.94E-01 |
| seed weight stem 1/dry weight stem 1 | median | enet | -0.18 | -0.09 | median | enet | -0.16 | -0.39 | BnaC09g50070D | -0.13 | 0.58 | 2.03E-01 |
| total seed weight/shoot dry weight | median | enet | -0.19 | -0.08 | median | enet | -0.17 | -0.39 | BnaA02g15500D | -0.12 | 0.33 | 1.81E-01 |
| end of shoot growth | hsic-5000 | rf | -0.22 | 0.02 | median | rf | -0.29 | -0.17 | BnaA05g09440D | 0.04 | 0.45 | 1.02E-02 |
| **Qualitative and low-count phenotypes** | **Feature sel.** | **Model type** | **Median test accuracy** | | **Feature sel.** | **Model type** | **Median test accuracy** | | | | | |
| leaf 8 lesions (76 DAS) | hsic-5000 | enet | 0.67 | | hsic-5000 | rf | 0.67 | | | | | |
| rosette lesions (74 DAS) | median | rf | 0.50 | | spearman | enet | 0.46 | | | | | |
| leaf 6 lesions (74 DAS) | median | enet | 0.33 | | spearman | enet | 0.50 | | | | | |
| stem count | median | enet | 0.33 | | hsic-5000 | enet | 0.50 | | | | | |

Table legend: Results are shown for models including all genes as potential features ('All genes' columns), models including only TFs as potential features ('Transcription factors' columns) and models using a single gene as feature ('Single gene' columns). For the best all-genes and TF models for continuous or high-count discrete phenotypes, columns from left to right indicate the feature selection technique used (median = selection of features with median *rlog* gene expression $> 0$, spearman = Spearman correlation, hsic-5000 = HSIC lasso, see Methods), the model type (enet or RF), the median test $R^2$ and the median pooled Pearson correlation coefficient (PCC, see Methods). Stars in the median test $R^2$ column indicate that the median test $R^2$ score on real data is higher than the 95th percentile of test $R^2$ scores on permuted data (S6 Data). For qualitative and low-count phenotypes, the median test accuracy was used as a performance metric instead of the median test $R^2$ (see Methods). Single-gene model columns include the best-performing gene and the corresponding median test $R^2$ and median pooled PCC. All single-gene models are cross-validated lme models with spatial error structure. The CV column contains the coefficients of variation for the phenotypes.

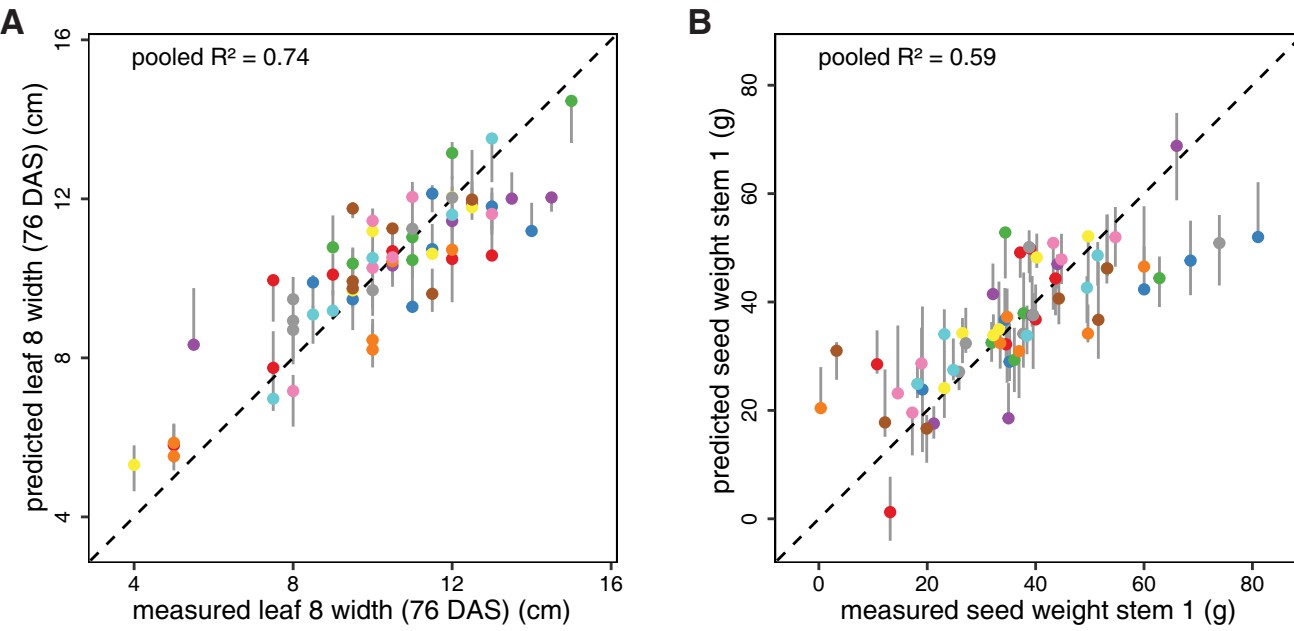

**Fig 2. Predictions versus observations for the best-scoring leaf and yield phenotypes. A.** Predicted versus measured values for leaf 8 width (76 DAS), using the all-genes model with the best median test $R^2$ score (enet + median feature selection, **Table 2**). **B.** Predicted versus measured values for seed weight stem 1, using the all-genes model with the best median test $R^2$ score (enet + Spearman feature selection, **Table 2**). Vertical grey lines range from the minimum to the maximum predicted value for a given plant across all model repeats, and colored dots represent predictions for the repeat with the median pooled $R^2$ score (i.e. the $R^2$ score of the pooled test set predictions in the repeat concerned). Different marker colors indicate the 10 different test sets in this repeat. Perfect predictions are located on the dashed diagonal line in each panel. Similar plots for other phenotypes are presented in **S6 Fig**.

on 90 datasets in which the phenotype values were permuted (**S6 Data**). In other words, the model for the real data train-test split with median test $R^2$ outperforms 95% of the models for comparable train-test splits on randomized data. Note that this serves only as an indication of model performance on real versus randomized data, not as a formal test assessing whether the median test $R^2$ score on real data is significantly higher than expected at random. The latter would require the 9 times repeated 10-fold cross-validation setup used on the real data to be used on each of the permuted datasets as well (instead of the single train-test split per permutation used here), which is computationally prohibitive.

Interestingly, seed count and weight, silique count and dry weight phenotypes measured for stem 1 are generally slightly more predictable than the corresponding phenotypes measured for the entire shoot, with median test $R^2$ score differences between stem 1 and total shoot phenotypes in the range [0.02, 0.09] for the all-genes models and [0.05, 0.13] for the TF models. This suggests that gene expression levels in leaf 8 of the rosette may be more predictive for phenotypes of stem 1 (i.e. the primary inflorescence stem and its cauline secondary inflorescences) than for phenotypes measured on the whole shoot (i.e. including the secondary inflorescence stems branching at ground level).

Root phenotypes, branching phenotypes, final plant height (278 DAS) and shoot growth phenotypes are generally poorly predictable (**Table 2**). Plant height and shoot growth phenotypes are likely poorly predictable because they show little variation across the field (**S3 Data**), increasing the risk that measurement error outweighs biological variation. Also taproot length and root system width may suffer from measurement errors. The total branch count and branch count stem 1 phenotypes on the other hand have a high CV and likely limited measurement error, suggesting that leaf 8 gene expression profiles may contain less information on these branching phenotypes than on leaf, seed, silique and dry weight phenotypes.

Most phenotypes calculated as ratios of other phenotypes are very poorly predictable, even if the constituent phenotypes have high prediction performance values. For instance, the median test $R^2$ value for seeds per silique (total seed count divided by total silique count) in the best all-genes model is negative (-0.14), whereas both total seed count and total silique count have median test $R^2$ values $\geq 0.38$ (**Table 2**). In many cases however, the numerator and denominator phenotypes of a ratio are highly correlated, leading to a derived phenotype with a small range that may be dominated by noise propagated from measurement errors in the constituent phenotypes rather than biological variability. The number of siliques per branch on stem 1 and the entire shoot are notable exceptions with high CV values and reasonable prediction performance in the best all-genes models (**Table 2**). The latter ratio phenotypes are highly correlated with the number of siliques on stem 1 (PCC = 0.92) and the entire shoot (PCC = 0.72), respectively, indicating that the number of siliques per branch is an important determinant of silique count, in addition to the number of branches (PCC between total branch count and total silique count = 0.87, PCC between branch count stem 1 and silique count stem 1 = 0.82).

The best all-genes model and best TF model generally have comparable performance for leaf 8 phenotypes, seed count and weight phenotypes, silique count phenotypes and branch count phenotypes, with median test $R^2$ differences (median test $R^2$ of best all-genes model— median test $R^2$ of best TF model) for these phenotypes in the range [-0.07, 0.08] (**Table 2**). In most of these cases, the median test $R^2$ score of the all-genes model is slightly higher than that of the TF model, but whether these differences are significant is hard to assess, as a commonly accepted framework for assessing the significance of out-of-sample $R^2$ differences between models is currently lacking. The $R^2$ differences between the best all-genes and TF models for dry weight phenotypes range from 0.05 to 0.10, suggesting that all-genes models may perform slightly better than TF models for these phenotypes. Among the other phenotypes for which either the best all-genes or best TF model has a median test $R^2$ score higher than the 95[th] percentile of test $R^2$ values obtained on permuted data (**S6 Data**, stars in **Table 2**), the best all-genes models for leaf 6 width (74 DAS) and siliques per branch stem 1 have a substantially higher median test $R^2$ score than the corresponding best TF models ($R^2$ difference = 0.31 and 0.23, respectively), while the best TF models for leaf count (74 DAS), rosette area (42 DAS) and root system width have a higher $R^2$ score than the corresponding best all-genes models ($R^2$ difference = -0.16, -0.13 and -0.16, respectively). In summary, the all-genes and TF models exhibit similar overall performance, in particular for the most predictable phenotypes. This indicates that, from the perspective of quantitative phenotype prediction, most of the information present in the complete gene expression dataset is also present in the TF gene expression data subset.

To compare multi-gene models, with either all genes or all TFs as potential features, to single-gene models in terms of phenotype prediction performance, we used the same repeated cross-validation setup as used for the multi-gene models to calculate median test $R^2$ scores and median pooled PCC values for single-gene models (lme models with spatial structure, see previous section). Cross-validation scores were calculated for each of the 100 genes most significantly associated with a given phenotype (lowest $q$-value for gene coefficient in lme model, **S4** and **S6 Data**).

For leaf 8 phenotypes, the best multi-gene models and best single-gene models have comparable median test $R^2$ scores ($\Delta R^2$ in the range [-0.05, 0.03], with $\Delta R^2$ = max(median test $R^2$ of best all-genes model, median test $R^2$ of best TF model)−median test $R^2$ of best single-gene model, **Table 2**). In other words, multi-gene models offer no benefit over single-gene models for quantitative prediction of leaf 8 phenotypes. Single- and multi-gene models also have comparable prediction performance for many of the yield traits measured on the entire shoot, such

as total seed weight, total seed count, total silique count and total dry weight with and without seeds ($\Delta R^2$ in the range [-0.05, 0.06], **Table 2**). There is also little difference in prediction performance for the dry weight of stem 1 with and without seeds ($\Delta R^2$ = 0.04 and 0.03, respectively). Many of the seed and silique traits related to stem 1 on the other hand (seed weight, seed count and silique count on stem 1, the number of siliques per branch on stem 1) are somewhat better predicted by multi-gene models than by single-gene models ($\Delta R^2$ in the range [0.09, 0.12], **Table 2**). This indicates that several distinct gene expression patterns may be relevant for quantitative prediction of stem 1 seed and silique traits.

For several other phenotypes, single-gene models outperform multi-gene models, sometimes with a wide margin, e.g. for plant height (278 DAS) ($\Delta R^2$ = -0.18), branch count on stem 1 ($\Delta R^2$ = -0.19) and leaf 6 length (74 DAS) ($\Delta R^2$ = -0.24). This suggests that the multi-gene models are vulnerable to overfitting. In particular phenotypes with low single-gene model performance tend to exhibit a multi-gene model performance that is even lower, suggesting that the extent of multi-gene model overfitting is inversely correlated with the proportion of trait variance explained by single genes. An alternative explanation for the observation that the best single-gene models sometimes outperform the corresponding multi-gene models may be the 'winner's curse' effect, also known as selection bias [36], whereby the apparently best-performing single-gene models may overestimate prediction performance.

For most ratio phenotypes, both the multi-gene and single-gene models have very poor prediction performance, in particular when the numerator and denominator phenotypes that make up the ratio are very highly correlated. In these cases, the denominator is essentially already a good predictor of the numerator. To assess whether any gene expression profiles contain additional information on the numerator given knowledge of the denominator, we used alternative single-gene models with a log link (see Methods) to predict the numerators of the seeds per silique ratio on stem 1 and the branches per stem ratio (seed count stem 1 and total branch count, respectively) conditioned on their denominator (silique count stem 1 and stem count, respectively). These models are not suited for making predictions in practice, given the need to know the denominator, but they may indicate whether prediction of the ratio based on gene expression is at all feasible and if so, which genes may be important. If no genes are found to be predictive for the numerator (and hence the ratio) conditioned on the denominator, then attempts to predict the ratio phenotype unconditionally are likely to be unsuccessful. For both seeds per silique stem 1 and branches per stem, the fitted coefficients and residuals look reasonable for the best predictor genes (**S7** and **S8 Figs**). The corresponding models succeed in suppressing a few of the more extreme residuals of the base model (without gene expression effect), without improving predictions for most other plants. However, no gene coefficients were found to be significantly different from zero for either phenotype after BH correction ($q \leq 0.05$), neither in models assuming constant error variance nor in models with heteroscedastic and/or spatially covarying error structures (see Methods). This indicates that the poor performance of the original multi-gene and single-gene models for these phenotypes is to be expected.

## Top predictors for leaf phenotypes

The best multi-gene prediction performance scores were obtained for leaf 8 phenotypes. To assess whether the genes featuring most prominently in the multi-gene models for leaf phenotypes make biological sense, we focused on the top-10 predictor lists of the TF-based models for leaf 8 length and width (76 DAS and 81 DAS), fresh weight (81 DAS) and area (81 DAS), and leaf 6 length and width (74 DAS) (**S6 Data**). As these leaf phenotypes are generally highly correlated (PCC between leaf 8 phenotype in the range [0.78,0.97], PCC between leaf 8 and

leaf 6 phenotypes in range [0.45, 0.60]), many of the most important predictors (TFs) in the random forest and elastic net models are shared among phenotypes. We therefore grouped the top-10 predictor lists for the different phenotypes in two sets, one for the RF models (**Fig 3**, n = 42) and one for the enet models (**S9 Fig**, n = 35). The rationale for looking at the TF models instead of the models built on all genes is that TFs are more likely than the average gene to have been functionally characterized to some extent, and are more likely to be causally involved in phenotype regulation (although it needs to be stressed that our analysis remains entirely correlational). Given the relative lack of experimentally determined gene functions in *B. napus*, most of the functional interpretation given below and in **S7 Data** for *B. napus* genes is based on experimentally determined functions of likely orthologs in *A. thaliana* (see Methods).

Many of the top TF predictors for leaf phenotypes have *A. thaliana* orthologs that have known functions in leaf development or exhibit overexpression, knockout or other mutant phenotypes related to leaf development (details in **S7 Data**). Both the RF and enet top predictor lists contain *BnaCnng05590D* and *BnaA05g33840D*, putative orthologs of the homeodomain leucine zipper class I (*HD-ZIP I*) gene *ARABIDOPSIS THALIANA HOMEOBOX 1* (*AtHB1/AT3G01470*). The enet top predictor list for leaf phenotypes also contains another *HD-ZIP I* gene, *BnaC02g43700D*, which is putatively orthologous to *AtHB5* (*AT5G65310*) or *AtHB16* (*AT4G40060*). Both *AtHB1* and *AtHB16* exhibit mutant phenotypes related to leaf development, seed yield and the timing of the vegetative-to-reproductive phase transition (**S7 Data**).

The RF and enet top predictor lists also contain several other *HD-ZIP* genes. *BnaA06g18550D* in the RF top predictor list is putatively orthologous to the *A. thaliana HD-ZIP III* gene *REVOLUTA* (*AtREV/AT5G60690*), which is involved in regulating postembryonic meristem initiation [37] and several polarity-associated growth processes in *A. thaliana*, including abaxial-adaxial patterning in leaves [38] (**S7 Data**). Additionally, *BnaC06g05240D* and *BnaA06g01940D* in the RF top predictor list are *HD-ZIP III* genes putatively orthologous to *AtHB8* (*AT4G32880*) or *AtHB15* (*AT1G52150*). *AtHB8* and *AtHB15* are thought to have effects on postembryonic meristem initiation that are antagonistic to the effects of *AtREV* [39], and to function prominently in vascular development, possibly antagonistically [40–43]. Furthermore, the enet top predictor list includes the *HD-ZIP II* gene *BnaC03g02700D*, putatively orthologous to *AtHAT3* (*AT3G60390*), *AtHAT14* (*AT5G06710*), *AtHB17* (*AT2G01430*) or *AtHB18* (*AT1G70920*). *AtHAT3* is known to be involved in leaf abaxial/adaxial patterning [44], and to be regulated by AtREV [45].

Both the RF and enet top predictor lists prominently feature putative orthologs of *A. thaliana WUSCHEL RELATED HOMEOBOX* (*AtWOX*) genes: *BnaA05g27750D* (RF and enet) and *BnaC05g41930D* (enet). Both genes are putatively orthologous to *AtWOX5* (*AT3G11260*) or *AtWOX7* (*AT5G05770*). Next to roles in root development, *AtWOX5* is known to also have functions in leaf development (**S7 Data**).

Not all transcription factors in the RF and enet models are equally important for all leaf phenotypes. *BnaCnng06440D* (*AtMYB60/AT1G08810*) for instance has higher RF and (to a lesser extent) enet importance scores for leaf 8 area (81 DAS) and leaf 8 fresh weight (81 DAS) than for other leaf phenotypes. Its likely *A. thaliana* ortholog *AtMYB60* is involved in regulating stomatal opening, and its expression is downregulated under drought [46] (**S7 Data**). A second TF in the RF models with higher importance for leaf 8 area (81 DAS) and leaf 8 fresh weight (81 DAS) than for other leaf phenotypes is *BnaC06g36000D* (*AtHB33/AtZHD5/ AT1G75240*). Its likely ortholog *AtHB33* codes for a zinc-finger homeodomain TF downregulated in response to abscisic acid (ABA), which e.g. induces stomatal closure [47]. A third TF in the RF models with mildly higher importance for leaf 8 area (81 DAS) and leaf 8 fresh

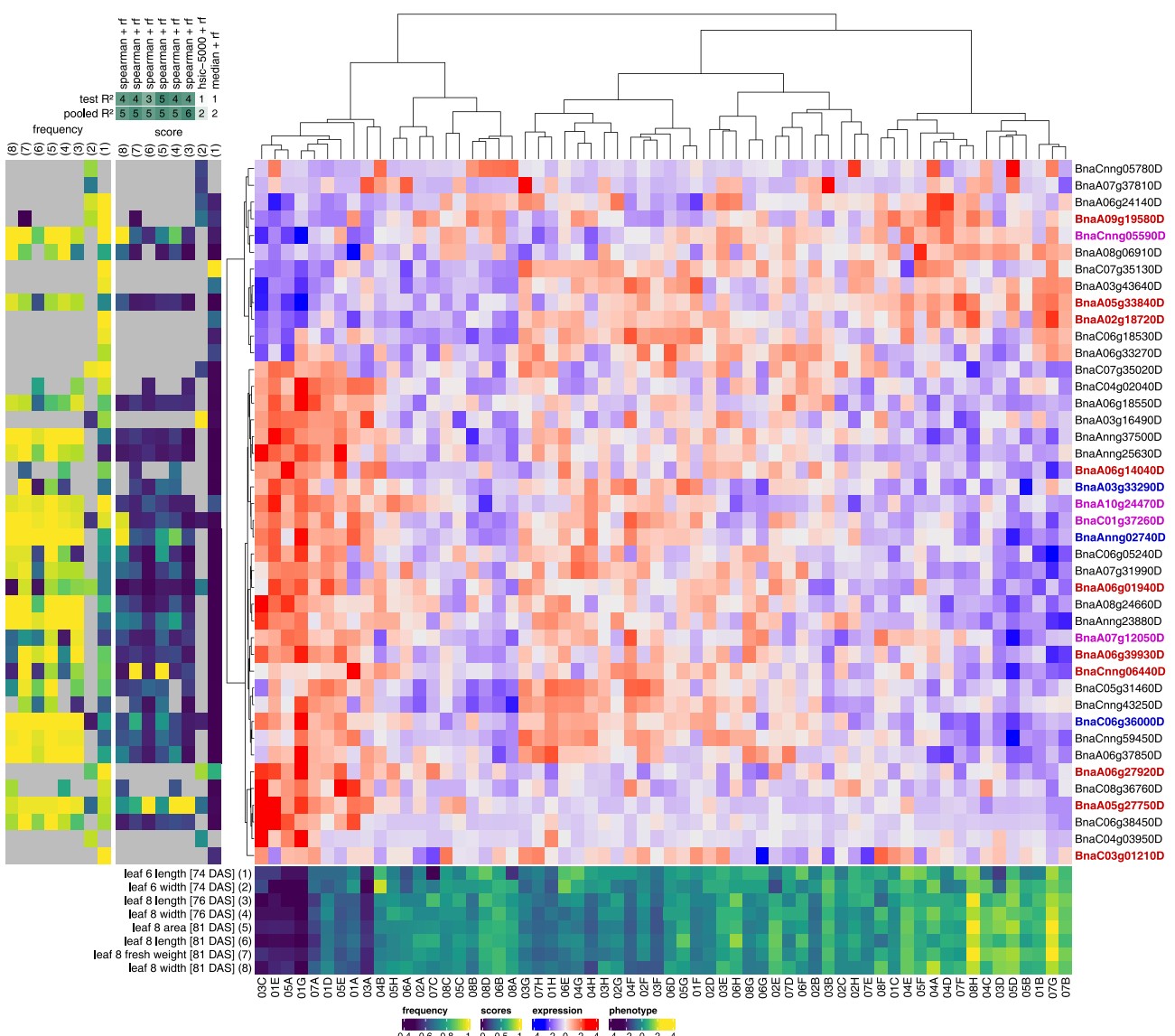

**Fig 3. Top predictor genes in RF models of leaf phenotypes.** A clustered heatmap of the z-scored gene expression profiles of the top genes for predicting leaf phenotypes is shown centrally (blue-red color scale, Ward.D2 hierarchical clustering). The leaf phenotypes concerned and their z-scored profiles across plants are shown at the bottom (dark blue-yellow heatmap with plant identifiers at the bottom). For each of these phenotypes, the top-10 most important genes (highest median gini importance across all 90 cross-validation splits) of the RF model with the highest median test $R^2$ score are included on the figure (gene identifiers are shown at right). The mostly dark blue score panel to the left of the expression heatmap shows the median gini importance scores of the selected genes in each of the selected phenotype models, normalized to the maximum importance score per model to make the color scales of the different models (columns) comparable. The mostly yellow frequency panel to the left of the score panel shows the frequencies at which genes were selected as features across all 90 cross-validation splits of a given model. Grey squares in the score and frequency panels indicate that a given gene was not selected as a feature in a given model. The phenotypes in the score and frequency panels are identified by numbers (1–8) on top of the panels, corresponding to the numbers associated with the phenotypes in the bottom phenotype panel. On top of the score panel, the feature selection techniques used in the best-scoring RF models for each phenotype are shown (median = selection of features with median *rlog* gene expression > 0, spearman = Spearman correlation, hsic-5000 = HSIC lasso, see Methods), as well as the corresponding test and pooled $R^2$ scores rounded to the nearest 0.1 and then multiplied by ten (e.g. a test $R^2$ score of 0.38 would be denoted as 4). Genes that are also found in the top-10 enet predictor lists for leaf phenotypes (**S9 Fig**) are highlighted in red, while genes that are also found in the top-10 enet or RF predictor lists for yield phenotypes (**Figs 4** and **S10**) are highlighted in blue. Genes found in both the top-10 enet predictor lists for leaf phenotypes and the top-10 enet or RF predictor lists for yield phenotypes are highlighted in magenta.

weight (81 DAS) is *BnaCnn59450D*, whose putative orthologs *AtSHN2* (*AT5G11190*) and *AtSHN3* (*AT5G25390*) have been linked to regulation of stomatal density and drought response [48] (**S7 Data**).

Both the RF and enet top-10 lists feature several orthologs of *A. thaliana NUCLEAR FACTOR Y, SUBUNIT A* (*AtNF-YA*) genes (putative *A. thaliana* orthologs in parentheses): *BnaAnng02740D* (*AtNF-YA2/10, AT3G05690/AT5G06510*, RF), *BnaA10g24470D* (*AtNF-YA2/10, AT3G05690/AT5G06510*, RF and enet), *BnaC06g33980D* (*AtNF-YA3/8, AT1G72830/AT1G17590*, enet) and *BnaC01g37260D* (*AtNF-YA5/6, AT1G54160/AT3G14020*, RF). All four genes are negatively correlated with leaf phenotypes in the field expression dataset (**S4 Data**). NF-Y transcription factor complexes are heterotrimers, consisting of A, B and C subunits, that function in various developmental programs and abiotic stress responses in plants [49]. Various *AtNF-YA* gene family members were previously found to function in the regulation of leaf size, drought resistance or the juvenile-to-adult (vegetative) phase change (**S7 Data**).

Interestingly, several of the top-TFs recovered in the multi-gene models for leaf phenotypes are linked to the regulation of flowering. Plant NF-Y complexes for instance are known to also function in the regulation of flowering time (**S7 Data**) [49]. It has been suggested that the photoperiodic flowering regulator CONSTANS (AtCO) may compete with NF-YA subunits in the NF-Y complex to form an alternative complex activating *FLOWERING LOCUS T* (*FT*) expression in *A. thaliana*, thereby promoting flowering [50]. *AtHB1*, *AtHB16* and *AtHB15*, the *A. thaliana* orthologs of several of the aforementioned *B. napus HD-ZIP* genes, have also been linked to regulation of the juvenile-to-adult and/or vegetative-to-reproductive phase changes [42, 51, 52]. Furthermore, both the enet and RF predictor lists contain *BnaA06g39930D*, a putative ortholog of *EARLY FLOWERING MYB PROTEIN* (*AtEFM/AT2G03500*) in *A. thaliana*. AtEFM is known to directly repress the expression of *FLOWERING LOCUS T* (*AtFT, AT1G65480*) in the leaf vasculature, and is thought to mediate the effects of temperature and light cues on the timing of the floral transition [53]. The RF predictor list additionally contains *BnaC05g31460D*, a putative ortholog of *AtJMJD5* (*AtJMJ30, AT3G20810*), the protein product of which interacts with AtEFM to repress AtFT [53]. The RF and enet top predictor lists also contain *BnaA07g12050D*, a putative ortholog of the floral homeotic gene *APETALA2* (*AtAP2/AT4G36920*) or the related *AtTOE3* (*AT5G67180*). Both AtAP2 and AtTOE3 are known to repress *AGAMOUS* (*AtAG*) expression during floral patterning [54].

In summary, 15/42 and 11/35 transcription factors in the RF and enet lists of top leaf phenotype predictors, respectively, have putative *A. thaliana* orthologs linked to leaf development and patterning, the juvenile-to-adult phase change, the floral transition or drought response (**S7 Data**).

## Top predictors for seed, silique and shoot dry weight phenotypes

Next to leaf 8 phenotypes, also the seed, silique and shoot dry weight phenotypes (further referred to as 'yield' phenotypes) of the individual plants at harvest (late spring) could be predicted to a considerable extent from autumnal leaf 8 transcriptome data (see above). Similar to the leaf phenotypes, the yield phenotypes are highly correlated (PCC range [0.84–0.99]) and hence have a lot of high-scoring RF and enet predictors in common (**S6 Data** and **Figs 4** and **S10**). Furthermore, these phenotypes are also significantly correlated with leaf phenotypes (PCC range [0.47, 0.74]), leading to a substantial overlap between the top-10 predictor lists of yield and leaf phenotypes.

In particular, virtually all TF genes in the leaf top-10 predictor lists with links to the juvenile-to-adult or vegetative-to-reproductive phase changes and flowering also feature prominently in the RF or enet top-10 predictor lists for yield phenotypes, including the *AtHB1* ortholog *BnaCnng05590D*, the *AtHB5/16* ortholog *BnaC02g43700D*, the *AtAP2* ortholog

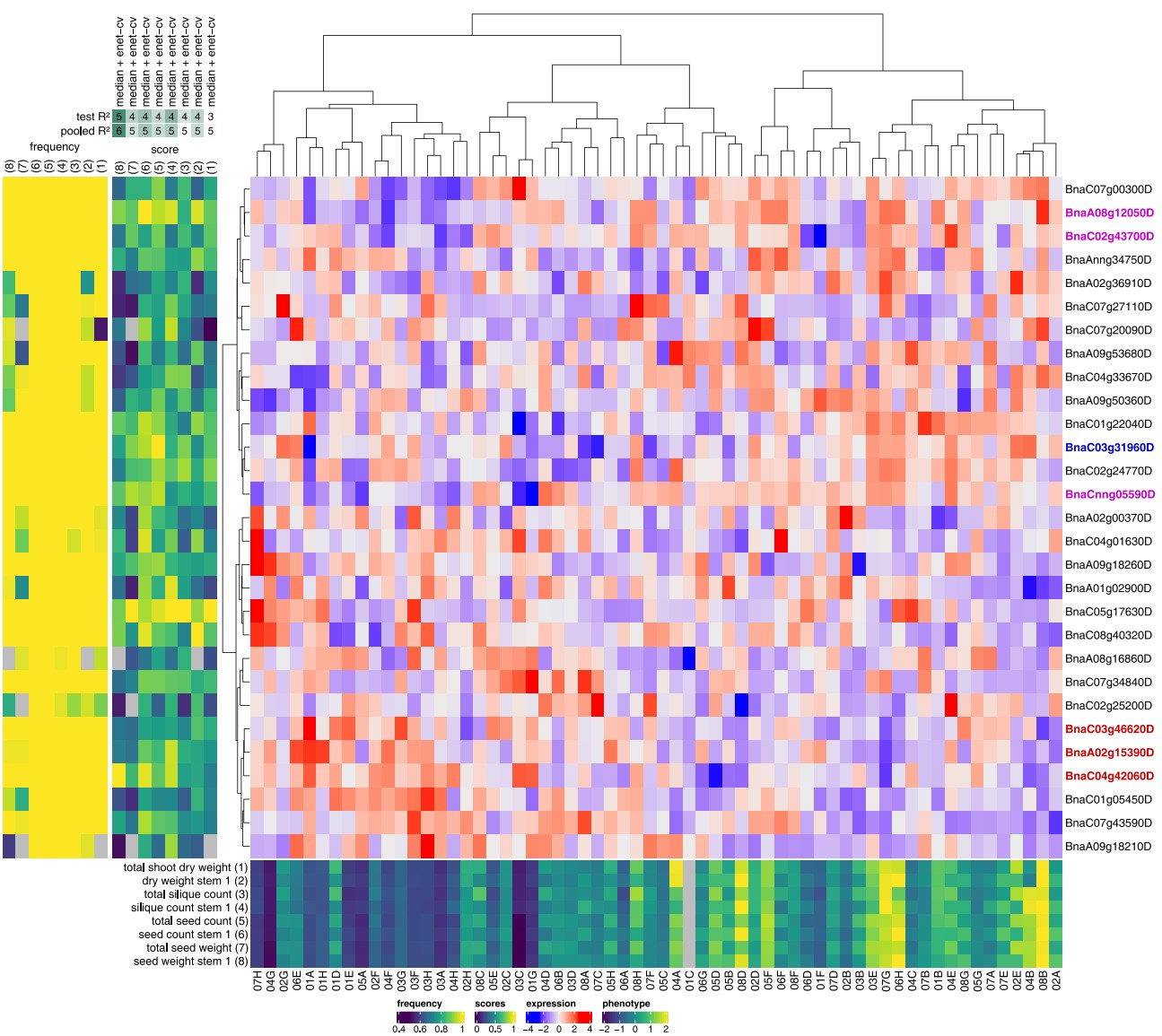

**Fig 4. Top predictor genes in enet models of yield phenotypes.** A clustered heatmap of the z-scored gene expression profiles of the top genes for predicting yield phenotypes is shown centrally (blue-red color scale, Ward.D2 hierarchical clustering). The yield phenotypes concerned and their z-scored profiles across plants are shown at the bottom (dark blue-yellow heatmap with plant identifiers at the bottom). For each of these phenotypes, the top-10 most important genes (highest median elastic net coefficients across all 90 cross-validation splits) of the enet model with the highest median test $R^2$ score are included on the figure (gene identifiers are shown at right). The mostly green-blue score panel to the left of the expression heatmap shows the median elastic net coefficients of the selected genes in each of the selected phenotype models, normalized to the maximum coefficient per model to make the color scales of the different models (columns) comparable. The mostly yellow frequency panel to the left of the score panel shows the frequencies at which genes were selected as features across all 90 cross-validation splits of a given model. Grey squares in the score and frequency panels indicate that a given gene was not selected as a feature in a given model. The phenotypes in the score and frequency panels are identified by numbers (1–8) on top of the panels, corresponding to the numbers associated with the phenotypes in the bottom phenotype panel. On top of the score panel, the feature selection techniques used in the best-scoring enet models for each phenotype are shown (median = selection of features with median *rlog* gene expression > 0, spearman = Spearman correlation, hsic-5000 = HSIC lasso, see Methods), as well as the corresponding test and pooled $R^2$ scores rounded to the nearest 0.1 and then multiplied by ten (e.g. a test $R^2$ score of 0.38 would be denoted as 4). Genes that are also found in the top-10 RF predictor lists for yield phenotypes (**S10 Fig**) are highlighted in red, while genes that are also found in the top-10 enet or RF predictor lists for leaf phenotypes (**Figs 3** and **S9**) are highlighted in blue. Genes found in both the top-10 RF predictor lists for yield phenotypes and the top-10 enet or RF predictor lists for leaf phenotypes are highlighted in magenta.

*BnaA07g12050D* and the *AtNF-YA* orthologs *BnaAnng02740D*, *BnaA10g24470D*, *BnaC06g33980D* and *BnaC01g37260D*. Furthermore, like the top predictor lists for leaf phenotypes, the enet top predictor list for yield phenotypes contains a putative ortholog of *AtEFM*, but a different one (*BnaAnng34750D*) (**S7 Data**).

Furthermore, many of the top predictor TF genes for yield phenotypes that are absent from the top-10 predictor lists for leaf phenotypes also have *A. thaliana* orthologs involved in processes related to the floral transition and flowering. In the combined set of top-10 enet predictors for shoot dry weight, seed and silique phenotypes (**Fig 4**, n = 29), five genes code for AGAMOUS-LIKE MADS-box transcription factors: *BnaC05g17630D* (*AtAGL104/ AT1G22130*), *BnaA02g15390D* (*AtAGL12/AT1G71692*), *BnaA02g00370D* (*BnFLC.A2*, *AtFLC/ AT5G10140*), *BnaA01g02900D* (*AtAGL16/AT3G57230*), and *BnaA09g53680D* (*AtAGL30/ AT2G03060*). *BnFLC.A2* is orthologous to *A. thaliana* FLOWERING LOCUS C (*AtFLC*), a key repressor of the floral transition [55, 56]. Two *AGAMOUS-LIKE* genes feature in the combined set of top-10 RF predictors for yield phenotypes (**S10 Fig**, n = 21): *BnaA02g15390D* (*AtAGL12/AT1G71692*) and *BnaA09g05500D* (*AtAGL8/AtFUL/FRUITFULL/AT5G60910*). *AtFUL* is thought to regulate the floral transition downstream of *AtFT* in the shoot apical meristem, partially redundantly with *AtSOC1* (*AtAGL20*, *AT2G45660*) [57, 58] (**S7 Data**).

The enet top predictor list also features *BnaA09g18260D*, a *HD-ZIP II* gene putatively orthologous to *JAIBA* (*AtJAB/AtHAT1/AT4G17460*) or *AtHAT2* (*AT5G47370*). *AtJAB* was shown to be involved in male and female reproductive development and floral meristem determination in *A. thaliana* [59] (**S7 Data**). The enet top predictor list also contains the *HD-ZIP IV* gene *BnaA09g50360D* (*AtHDG2/AT1G05230*). A combination of *hdg2* and *pdf2* null mutant alleles in *A. thaliana* was shown earlier to produce flowers with sepaloid petals and carpeloid stamens [60].

Furthermore, the gene *BnaA08g12050D* is ranked highly in both the enet and RF top predictor lists. The best candidate ortholog of this gene in *A. thaliana* is *AtMYB3R1* (*AT4G32730*), coding for a regulator of cell proliferation that acts in a module with AtTSO1 to balance cell proliferation with differentiation in developing roots and shoots [61]. *BnaA08g12050D* also features as a predictor for leaf 6 length (74 DAS) and leaf 8 area (81 DAS) in **S9 Fig**.

*BnaC07g27110D* and *BnaC01g22040D* in the enet predictor list are putative orthologs of *AtGATA16* (*AT5G49300*) and *AtGATA17* (*AT3G16870*) or *AtGATA17L* (*AT4G16141*), respectively. Evidence suggests these and other LLM-domain B-GATA transcription factors are involved (at least partially redundantly) in the regulation of flowering time, silique length, seed set and other developmental processes [62]. The enet top predictor list also contains *BnaC04g33670D* and *BnaA08g16860D*, *BZIP* genes putatively orthologous to the *A. thaliana* genes *DRINK ME* (*AtDKM/AtBZIP30/AT2G21230*) and *DRINK ME-LIKE* (*AtDKML/AtB-ZIP29/AT4G38900*), respectively. *AtDKM* and *AtDKML* are negative regulators of reproductive development and growth (**S7 Data**). AtDKM was shown to interact *in planta* with several regulators of meristem development, including WUSCHEL (AtWU), HECATE1 (AtHEC1), the aforementioned JAIBA and NGATHA1 (AtNGA1) [63]. Interestingly, the RF top predictor list contains a putative ortholog of *NGATHA1* (*AtNGA1/AT2G46870*) or *NGATHA2* (*AtNGA2/AT3G61970*), namely *BnaA09g39540D*. *AtNGA1* and *AtNGA2* are known to be involved in gynoecium development [64, 65] and were recently shown to also have a function in regulating shoot apical meristem development [66]. Another likely regulator of meristem development, *BnaC07g43590D*, is found in the enet predictor list. *BnaC07g43590D* is most likely an ortholog of *ARABIDOPSIS RESPONSE REGULATOR 10* (*AtARR10/AT4G31920*) or *12* (*AtARR12/AT2G25180*), both known to directly activate the expression of *WUSCHEL* and to play a role in shoot apical meristem regeneration and maintenance [67].

In summary, 16/29 and 11/21 TF genes in the enet and RF lists of top yield predictors, respectively, have putative *A. thaliana* orthologs linked to the juvenile-to-adult phase change, the floral transition, flowering or regulation of meristem development.

## Predicting final yield phenotypes from early growth phenotypes

As a baseline to assess the prediction performance of the molecular models, we trained models predicting plant phenotypes in spring (mostly phenotypes at harvest) from single or multiple autumnal leaf and rosette phenotypes. For these single- and multi-phenotype models, the same modeling approaches were used as for the single- or multi-gene models, respectively (see Methods).

Interestingly, many of the mature plant phenotypes can be predicted to a considerable extent from phenotypes measured earlier in the growing season (**Table 3** and **S8 Data**). In particular the models for phenotypes measured on the entire shoot (total seed, silique and branch count, total seed weight, total shoot dry weight) perform surprisingly well. For most of these

**Table 3. Best-performing multi-phenotype and single-phenotype models for mature plant phenotypes.**

| Mature plant phenotypes | All early phenotypes | | | Single early phenotypes | | |
|---|---|---|---|---|---|---|
| | Model type | Median test R2 | Median pooled PCC | Top phenotype | Median test R2 | Median pooled PCC |
| seed weight stem 1 | enet | 0.33 | 0.59 | leaf 8 area (81 DAS) | 0.32 | 0.63 |
| seed count stem 1 | rf | 0.25 | 0.57 | leaf 8 area (81 DAS) | 0.26 | 0.61 |
| silique count stem 1 | enet | 0.24 | 0.55 | leaf 6 width (74 DAS) | 0.22 | 0.53 |
| total seed count | rf | 0.40 | 0.66 | leaf 8 area (81 DAS) | 0.44 | 0.71 |
| dry weight stem 1 | enet | 0.26 | 0.57 | leaf 8 area (81 DAS) | 0.35 | 0.62 |
| dry weight stem 1 (w/o seeds) | enet | 0.18 | 0.54 | leaf 8 area (81 DAS) | 0.30 | 0.59 |
| total seed weight | enet | 0.45 | 0.68 | leaf 8 area (81 DAS) | 0.46 | 0.72 |
| total shoot dry weight | enet | 0.38 | 0.67 | leaf 8 area (81 DAS) | 0.44 | 0.71 |
| total silique count | rf | 0.36 | 0.63 | leaf 8 area (81 DAS) | 0.41 | 0.70 |
| siliques per branch stem 1 | enet | 0.14 | 0.47 | leaf 8 area (81 DAS) | 0.07 | 0.48 |
| total shoot dry weight (w/o seeds) | enet | 0.29 | 0.63 | leaf 8 area (81 DAS) | 0.37 | 0.68 |
| branch count stem 1 | enet | 0.35 | 0.64 | leaf 8 area (81 DAS) | 0.34 | 0.65 |
| siliques per branch | enet | -0.04 | 0.32 | leaf 6 width (74 DAS) | -0.07 | 0.36 |
| plant height | enet | 0.23 | 0.58 | leaf 8 length (81 DAS) | 0.25 | 0.61 |
| total branch count | rf | 0.40 | 0.69 | rosette area (42 DAS) | 0.38 | 0.65 |
| branch count stem 1/length stem 1 | rf | 0.33 | 0.63 | leaf 8 area (81 DAS) | 0.22 | 0.56 |
| max shoot growth rate | enet | 0.04 | 0.40 | leaf 8 width (81 DAS) | 0.04 | 0.41 |
| root system width | rf | 0.04 | 0.42 | leaf 8 length (81 DAS) | 0.05 | 0.39 |
| time of max shoot growth | enet | -0.01 | 0.53 | leaf 8 width (81 DAS) | 0.08 | 0.53 |
| taproot length | rf | -0.02 | 0.33 | leaf 8 width (81 DAS) | 0.01 | 0.33 |
| branches per stem | enet | -0.14 | -0.22 | leaf 8 lesions (76 DAS) | -0.15 | 0.14 |
| seeds per silique | enet | -0.17 | -0.18 | leaf 8 length (81 DAS) | -0.07 | 0.18 |
| seeds per silique stem 1 | enet | -0.15 | -0.06 | leaf 8 length (81 DAS) | -0.04 | 0.23 |
| seed weight stem 1/dry weight stem 1 | enet | -0.15 | -0.40 | leaf 8 lesions (76 DAS) | -0.19 | -0.19 |
| total seed weight/shoot dry weight | enet | -0.16 | -0.39 | leaf 8 lesions (76 DAS) | -0.18 | -0.06 |
| end of shoot growth | enet | -0.15 | 0.20 | leaf 8 width (81 DAS) | -0.12 | 0.30 |

Table legend: Results are shown for models including all early phenotypes as potential features (multi-phenotype models) and models using a single early phenotype as feature (single-phenotype models). For the best multi-phenotype models, columns from left to right indicate the model type used (enet or RF), the median test $R^2$ and the median pooled PCC (see Methods). Single-phenotype columns include the best-performing early phenotype ('Top phenotype' column) and the corresponding median test $R^2$ and median pooled PCC. All single-phenotype models are cross-validated lme models with spatial error structure.

phenotypes, the performance of the early-phenotype models is only slightly less than that of the best single-gene or multi-gene model, and the early-phenotype models for total seed weight and total branch count even outperform the molecular models ($\Delta R^2$ = −0.04 and −0.14, with $\Delta R^2$ = max(median test $R^2$ of best multi-gene model, median test $R^2$ of best single-gene model) −max(median test $R^2$ of best multi-phenotype model, median test $R^2$ of best single-phenotype model), **Tables 2** and **3**). Also for branching phenotypes related to stem 1 (branch count stem 1, branch count stem 1/length stem 1), the best early-phenotype models feature high prediction performance scores. For other stem 1 phenotypes however (seed weight, seed count, silique count and siliques per branch on stem 1, stem 1 dry weight with and without seeds), the molecular models clearly outperform the early-phenotype models ($\Delta R^2$ in range [0.09, 0.23], **Tables 2** and **3**).

Most multi-phenotype models with appreciable prediction performance (median test $R^2$ > 0.10), both for whole-shoot and stem 1 phenotypes, feature leaf 8 area (81 DAS) as the top predictor (**Table 3**). Leaf 8 area (81 DAS) is generally also the most predictive early phenotype in the corresponding sets of single-phenotype models. The multi-phenotype models with the best prediction performance scores, i.e. those for whole-shoot phenotypes and stem 1 branching phenotypes (but not the other stem 1 phenotypes), generally also feature rosette area (42 DAS) as a predictor of some importance (**S8 Data**). For total branch count, branch count stem 1 and branch count stem 1/length stem 1, rosette area (42 DAS) is even the top predictor in either the RF or enet model, or both (**S8 Data**). Rosette area (42 DAS) itself is only moderately predictable from the leaf 8 molecular data, which may explain why multi-phenotype models are better at predicting these branching phenotypes than multi-gene models.

Our results indicate that the leaf 8 molecular data offer little benefit over early-phenotype measurements for quantitative prediction of mature phenotypes measured on the entire plant. On the other hand, the leaf 8 molecular data yields substantially better models than the early-phenotype data for most mature stem 1 phenotypes. Often, the multi-gene models for stem 1 phenotypes are also slightly better than the multi-gene models for the corresponding whole-plant phenotypes (see previous section). This suggests that the molecular makeup of the 8th rosette leaf at the time of sampling contained more information on the development of the primary flowering stem and its cauline secondary inflorescences than on the development of side stems at ground level. Early phenotypes on the other hand may contain more information on whole-plant yield phenotypes than on phenotypes specifically related to stem 1.

Given that even the earliest of the autumnal phenotypes considered thus far, the rosette area at 42 DAS, still has some predictive power for several yield phenotypes (median test $R^2$ > 0.10 for total branch count, seed count, seed weight and silique count, total dry weight with and without seeds, branch count stem 1 and branch count stem 1/length stem 1), we assessed whether earlier rosette areas (v2, see Methods) are also predictive for these phenotypes (**Fig 5** and **S9 Data**). Median test $R^2$ scores were found to decrease when using earlier rosette areas as predictors, with rosette areas measured $\leq$ 28 DAS generally yielding low (< 0.10) and in many cases negative median test $R^2$ scores. When using the earliest rosette area (14 DAS) as predictor, the median pooled $R^2$ and PCC scores are however still in the ranges [0.05, 0.20] and [0.27, 0.45], respectively, indicating that even the earliest rosette area measurements contain some information on final yield phenotypes (**S9 Data**).

## Discussion

In this study, we used machine learning models to predict the phenotypes of individual *B. napus* Darmor plants grown in the same field from rosette-stage leaf gene expression data. Our results show that many plant phenotypes can be predicted to a substantial extent from leaf 8

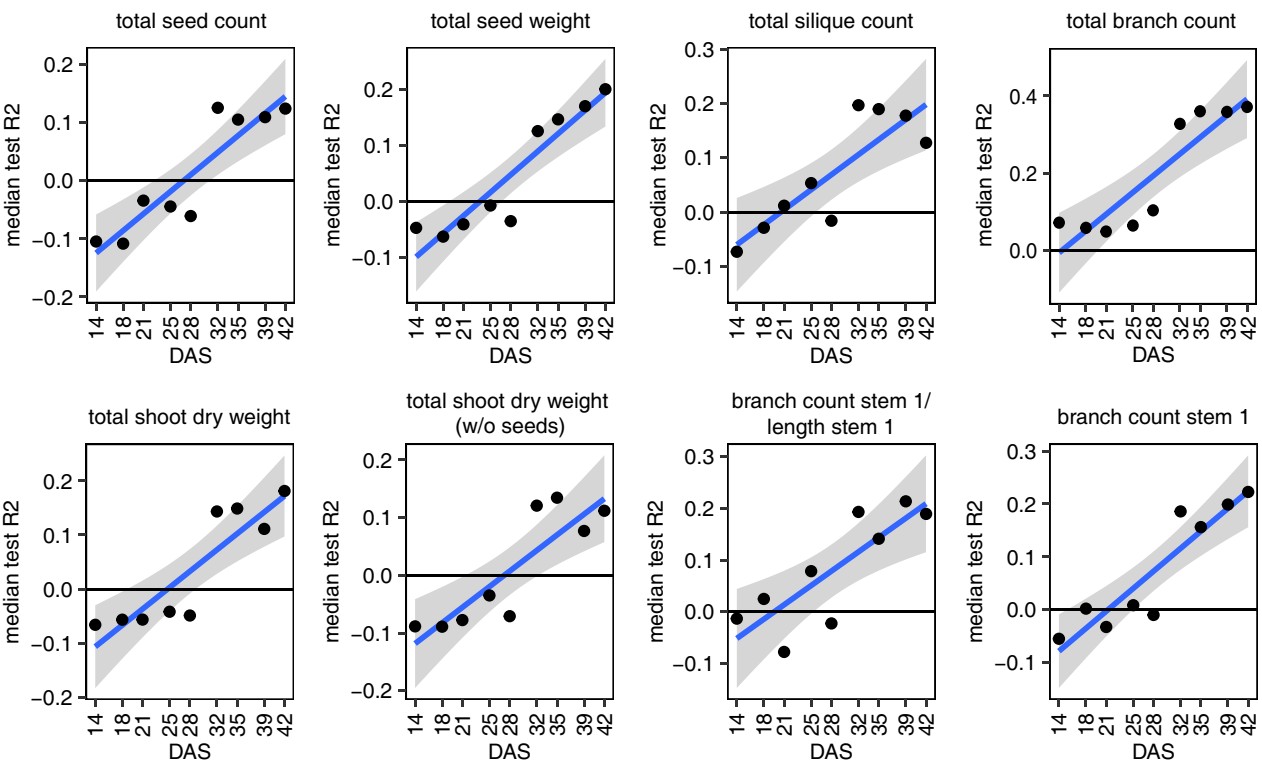

**Fig 5. Predictive power of early rosette areas for yield phenotypes.** In each subplot, median test $R^2$ values are plotted for lme models predicting the given phenotype from early rosette areas v2 (14–42 DAS, x-axis). Only mature phenotypes that can be predicted from rosette area (42 DAS) with a median test $R^2 > 0.1$ are shown. Blue lines are ordinary least-squares linear regressions, with shaded areas indicating 95% confidence intervals on the trendline. Most phenotypes exhibit a rather dichotomous median test $R^2$ profile with rosette areas v2 from 14 to 28 DAS yielding substantially lower median test $R^2$ values than rosette areas v2 from 32 to 42 DAS. Accordingly, linear model fits at 28 and 32 DAS are often poor.

gene expression. Phenotypes closely related in time and space to the material sampled for RNA-seq, in particular leaf 8 phenotypes, generally feature good prediction performance, in accordance with results obtained earlier in a similar setup for maize [28]. Interestingly however, also many of the phenotypes measured at the end of the growing season, ~5.5 months after leaf sampling for RNA-seq, feature high prediction performance. In particular seed yield, silique and dry weight traits exhibit prediction performance scores in the same range as the autumnal leaf and rosette phenotypes.

Azodi *et al.* [68] predicted several agronomically relevant mature plant traits (plant height, grain yield and flowering time) in a population of maize inbred lines from genetic marker data, whole-seedling transcriptome data and combinations thereof. Their transcriptome-based models exhibited PCC scores between predicted and measured values in the range [0.50, 0.61] for flowering time, [0.42, 0.51] for plant height and [0.47, 0.55] for 300 kernel weight [68]. In the present study, the transcriptome-based (all-genes) models for mature plant traits in *B. napus* (ignoring ratio phenotypes) exhibit median pooled PCC scores in the range [0.57, 0.77] for seed phenotypes, [0.51, 0.74] for silique phenotypes, [0.56, 0.73] for shoot dry weight phenotypes, [0.40, 0.56] for branch count phenotypes, [0.40, 0.53] for plant height (278 DAS) and [0.07, 0.36] for root phenotypes (**Table 2** and **S6 Data**). Comparing the observed PCC ranges of both studies suggests that mature traits of individual plants of the same line grown in the same field are as predictable from early-stage transcriptome data as average mature traits in a diversity panel. However, direct comparison of the PCC values across studies is complicated

by differences in the phenotypes predicted, prediction and scoring methodology and factors affecting model training, scoring and overfitting potential such as the study population size (388 lines in the maize study versus 62 *B. napus* plants in the present study) and the number of potential model features (31,238 genes in the maize study versus 76,808 genes in the *B. napus* dataset). Also the species difference and the tissue and developmental time point sampled for RNA-seq (whole seedlings at the V1 stage for maize versus rosette leaf 8, 81 DAS, for *B. napus*) may impact how well a transcriptome can predict a given phenotype. The most comparable models are likely the whole-transcriptome-based random forest model for maize plant height, with a PCC of 0.42 [68], and the median-filter all-genes random forest model for the height of individual *B. napus* plants (without feature selection other than removing genes with *rlog* expression > 0 in less than half of the samples, reducing the feature set to 55,166 genes), with a median pooled PCC of 0.43 (**S6 Data**).

Given that the single-plant transcriptome data can quantitatively predict many plant phenotypes better than expected by chance, the top predictor genes may shed light on biological processes that impact phenotypes in the field. Many of the top predictors in the TF models for seed, silique and dry weight phenotypes for instance are known to function in the floral transition. From the perspective of our experimental setup, it makes sense that such genes are recovered, as it is known that the floral transition starts in autumn in winter-type *B. napus* accessions [69, 70], i.e. around the time that rosette leaves were harvested for RNA-seq in the present field trial, and is set in motion to a large extent by systemic signals emanating from leaves in Brassicaceae and other plant families [71–73].

Mechanistic interpretation of the correlational links between top predictor genes and phenotypes is however not straightforward. Putative orthologs of *AtHB1* and *AtHB16* are for instance found among the top predictors positively correlated with both leaf and yield phenotypes (**Figs 4** and **S10** and **S4 Data**), but upregulation of these genes in *A. thaliana* was previously found to lead to smaller and more serrated leaves [52, 74], to delay the vegetative-to-reproductive phase transition and to result in siliques bearing fewer seeds [51, 52]. Some top predictors that correlate negatively with yield phenotypes have putative *A. thaliana* orthologs that are thought to function primarily as negative regulators of the floral transition in leaves, e.g. *AtNF-YA* genes [50, 75], but others are putatively orthologous to a positive regulator of the floral transition, such as *AtFUL*. Other floral transition regulators recovered as predictors in our yield models, e.g. orthologs of *AtFLC* and *AtEFM*, do not by themselves exhibit a significant positive or negative correlation with yield phenotypes.

Most likely, the associations recovered between individual plant phenotypes and autumnal leaf gene expression patterns are due to developmental timing differences among the plants, rather than reflecting the effects of upregulation or downregulation of specific regulators. In the *A. thaliana* developmental gene expression atlas of Klepikova *et al.* [76], orthologs of predictors positively correlated with leaf size such as *AtHB1* and *AtHB16* (**S4 Data**) are more highly expressed in mature *A. thaliana* leaves (at flowering), while orthologs of predictors negatively correlated with leaf size such as *AtREV*, *AtWOX5* and *AtHAT3* are more highly expressed in young leaves. This suggests that plants with low expression of *AtHB1/16* orthologs and high expression of *AtREV/AtWOX5/AtHAT3* orthologs had a more juvenile (and hence smaller) leaf 8 at the autumnal sampling time point, which explains the observed gene expression-leaf phenotype correlations. That autumnal leaf phenotypes and final yield phenotypes have several developmental predictors in common (e.g. *AtHB1*) and that the autumnal leaf phenotypes themselves are also predictive of yield indicates that the developmental differences in autumn impacted final yield. These differences were not limited to differences in leaf development, as evidenced by the fact that the predictor sets for both leaf and yield phenotypes also

contain regulators of plant-wide developmental phase transitions occurring in autumn (juvenile-to-adult, vegetative-to-reproductive).

In summary, our results indicate that the yield potential of the individual plants was already determined to a large extent by their developmental state at the time of leaf sampling in autumn. Mendham and Scott [77] previously found that the size of winter-type *B. napus* plants at the time of inflorescence initiation affects their yield potential, in the context of an experiment assessing sowing date effects on yield. Our results show that even when sown on the same date in the same field, individual winter-type *B. napus* plants of the same line display developmental differences in autumn that correlate with yield differences in spring. Even if only part of the variability in e.g. total seed weight (CV = 46.9%) observed in our trial is due to autumnal effects on plant growth and development, the gains of mitigating such effects could be substantial.

The question remains however what could have caused the developmental differences among plants in the present field trial. One potential cause is differences in seed germination and seedling emergence across the field. In wheat, it was established previously that relative differences in seedling emergence date are strongly correlated with differences in final yield [78]. Next to seed quality, many environmental factors are known to impact the timing of seed germination and seedling emergence, including soil structure [79], soil temperature [80], sowing depth [78, 80], soil water potential, oxygenation and light quality [81], and soil nutrients [82]. The seedling emergence date was not recorded in the present field trial, but the closest proxy that was measured, namely rosette area at 14 DAS, was found to be a bad predictor for yield (**Fig 5** and **S9 Data**), indicating that variation in seed germination and seedling emergence across the field did not by themselves have a major impact on yield in the present trial. The observation that later rosette areas are progressively better at predicting yield rather suggests that developmental differences among plants accumulated over time. It should be noted however that the variation in seedling emergence in the present trial was mitigated by preferential pruning of early- and late-emerging seedlings at every grid position (see Methods), rendering our trial unsuited to assess the effects of seedling emergence on yield in general.

The observation that genes involved in the regulation of circadian rhythm, photoperiodism and the vegetative-to-reproductive phase transition are on average more spatially autocorrelated in the autumnal gene expression dataset than the average gene suggests that spatially patterned micro-environmental factors may be linked to the variability of developmental gene expression in autumn, and ultimately yield variability in spring. That the phenotypes are influenced by environmental factors is also suggested by the observation that the sets of genes associated with leaf and yield phenotypes are heavily enriched in genes involved in responses to abiotic and biotic stimuli and nutrient levels (**S5 Data**). The finding that developmental processes feature more prominently in the TF-based phenotype prediction models than responses to environmental stimuli indicates that micro-environmental variations among plants in the present field trial may have influenced plant phenotypes mainly by influencing development. More work is needed however to establish whether and how micro-environmental variability impacts the growth and development of individual plants in the same field. To address this, additional field trials need to be performed in which, next to the gene expression and phenotypes of individual plants, also a range of environmental parameters is measured on the single-plant level (e.g. soil structure and chemistry).

Additional single-plant field trials are also needed to assess to what extent the predictive models, gene-phenotype and process-phenotype associations learned from the present field trial generalize to other soils and meteorological conditions, other time points or tissues sampled for RNA-seq, and other cultivars. Given the developmental nature of many of the top predictors in the current models, it is likely that our current prediction models, based on leaf gene

expression data for a single field trial at a single time point, will not perform well when applied on follow-up field trials, even when using the same cultivar in a similar field under roughly the same climate conditions. Differences in weather conditions and other environmental factors across trials may for instance influence the timing of developmental phase transitions, making it all but impossible to sample the exact same developmental time window in follow-up trials. If leaf gene expression were to be profiled at a slightly earlier or later developmental time window than in the present trial, the current top predictors may no longer be adequate phenotype proxies and other genes that function earlier or later in e.g. the floral transition may become relevant instead. The construction of robust prediction models will therefore likely require single-plant data generated under a wide variety of field conditions and sampling schemes. We want to emphasize however that quantitative prediction of single-plant phenotypes is not the primary goal we envision for single-plant omics experiments. Rather, the primary aim is to identify which biological processes, environmental factors and associated genes may influence plant phenotypes in the field. In this respect, any additional genes and processes identified in follow-up trials would add to our overall knowledge on how rapeseed plants grow in a field.

It is worth pointing out that the dataset generated in this study may also serve other purposes than gene-phenotype association. Earlier, we have shown that field-generated single-plant transcriptomics data can also be used efficiently to predict the function of genes [28]. Given the complex genome duplication history of *B. napus* [83], the combination of gene function prediction and gene-phenotype association may be particularly useful to shed light on which *B. napus* genes in a (long) list of paralogs are most likely functionally orthologous to a given *A. thaliana* gene, and how paralogs have diverged in function. This knowledge may in turn be useful in the context of genetic engineering and breeding efforts to optimize yield and stress tolerance in *B. napus*.

## Materials and methods

### Field trial setup

Seeds from the winter-type *Brassica napus* accession Darmor (BnASSYST-120) were sown in a field in Merelbeke, Belgium (50˚58'24.9"N 3˚46'49.1"E) on September 8, 2016. Three seeds were sown at ~2 cm depth at each of 100 points arranged in a 10x10 grid with 0.5 m spacing within and between rows. Seedlings were thinned out to leave one seedling growing at each grid point. Early- and late-emerging seedlings were pruned preferentially (based on visual assessment) to make the remaining seedling population as homogeneous as possible. At two points, no seedlings emerged.

Plots of *Miscanthus sinensis*, *M. sacchariflorus* and *Miscanthus* hybrids were grown to the northeast and southeast of the *B. napus* field trial, and maize was grown to the northwest, at distances > 5 m. The field plot was surrounded by chicken wire and covered by netting to keep out birds and large herbivores. The netting was removed in spring when plants grew taller than ~1 m. Additionally, perimeter fencing was used to protect the field trial and the mobile weather station on site (see **S1 Data** for weather station data).

After germination, individual plant images were taken twice a week between September 22 and October 20, 2016 (9 time points) to assess the projected leaf area of the growing rosettes. Nadir images were taken using a D90 camera (Nikon Inc., USA) equipped with a 35 mm lens (AF-S DX Nikkor 35 mm F1.8G, Nikon Inc., USA) set at iso 200, f/8. The shutter speed could vary to allow for a proper exposure, determined by the camera. The camera/tripod was positioned away from the sun to avoid shadows in the images taken. For each time point a grey calibration card (Novoflex grey card 15 x 20 cm, NOVOFLEX Präzisionstechnik GmbH, Germany) was used to correct the white balance. This card was also used as reference to

convert pixels to areas in cm$^2$ (see below). The ground sampling difference (GSD) was 0.015 cm/pixel.

At 74 DAS, the length and width of leaf 6 (counting upward from the first true leaf) were measured non-destructively, leaf 6 lesion and total rosette lesion severity were scored and the number of fully emerged rosette leaves (area > ~2 cm$^2$) was recorded. At 76 DAS, leaf 8 length and width were measured and leaf 8 lesions were scored. The width of leaf blades was measured at the widest point. Leaf lengths were measured from the leaf tip to the point where the petiole first lacked conspicuous laminar tissue (looking from the leaf tip toward the base). Lesion severity was scored qualitatively on a scale from 0 (lesions cover at most five percent of the leaf blade or rosette) to 2 (more than half of the leaf or rosette eaten).

At 81 DAS, on November 28, 2016, the eighth rosette leaves of 62 non-border plants (i.e. the plants at all non-border locations where seedlings emerged) were harvested for RNA-sequencing in a time span of ~1 hour (13:25–14:27). Leaves were cut off where the petiole first lacked conspicuous laminar tissue (looking from the leaf tip toward the base) and washed with DEPC-treated and sterilized water. The chlorophyll content of each leaf was measured at four different positions on the leaf with a CCM-200 chlorophyll content meter (Opti-Sciences, Inc., Hudson, USA), and the average of these measurements was used in the analyses. Leaves were then photographed twice against a white background with a piece of millimeter paper to assess the image scale and perspective, a ruler, and color and greyscale references, the second time covered with a glass plate to flatten them. Next, the midvein of every leaf was cut out using scissors, and the residual leaf material was folded into a pre-weighed 50 ml tube. The filled 50 ml tube and the midvein were weighed together to measure leaf fresh weight, after which the tube was stored in liquid nitrogen on the field. The entire leaf processing pipeline, from cutting a leaf to storing it in liquid nitrogen, was completed for each leaf in less than 5 minutes.

After leaf sampling, the plants were left to overwinter and set seed in spring. After bolting, plant height was measured from ground level to the top of the primary flowering stem at 13 time points between 189 and 231 DAS (S1 Data). One of the plants sampled in autumn for RNA-seq, 01C, did not survive until the end of the growth season. The remaining 61 non-border plants were harvested on June 13, 2017 (278 DAS), at which time ~50% of seeds had started changing color from green to black but no significant pod shattering or seed predation had occurred. Final plant height at 278 DAS was measured on the field, from ground level to the top of the primary flowering stem. Afterwards, shoots were cut off and the root systems were dug up. Taproot length was measured from ground level to the deepest root tip. Root system width was measured perpendicular to the taproot at the root system's widest point.

For each harvested plant, the primary flowering stem plus its cauline secondary inflorescences (stem 1) and the secondary inflorescence stems branching at ground level (side stems) were dried in two separate bags in a well-ventilated, dry attic. The number of branches and siliques per stem, the total shoot dry weight and the dry weight of stem 1 were measured on dried plants. Seeds were recovered manually from the dried-out pods for stem 1 and the side stems separately, and separated from dust and small pod debris using a customized seed aspirator with vibration channel (Baumann Saatzuchtbedarf GmbH, Waldenburg, Germany). The resulting seed batches for stem 1 and the side stems were weighed and counted using an elmor C3 seed counter (elmor AG, Schwyz, Switzerland). Seed counts and weights are reported for stem 1 and the entire plant (i.e. the sum of stem 1 and the side stems).

## Determination of shoot growth parameters

Shoot growth parameters (time of maximum shoot growth $t_m$, maximum shoot growth rate and the end of shoot growth $t_e$) were derived by fitting a beta-sigmoid growth curve to the

time series of 14 plant height measurements between 189 and 278 DAS [84]:

$$h(t) = h_0 + (h_{max} - h_0) * \left(1 + \frac{t_e - t}{t_e - t_m}\right) * \left(\frac{t}{t_e}\right)^{\frac{t_e}{t_e - t_m}} \quad t < t_e$$

$$h(t) = h_{max} \qquad\qquad\qquad\qquad\qquad\qquad t \geq t_e \qquad (\text{Eq1})$$

With $h(t)$ the plant height at plant age $t$, $h_0$ and $h_{max}$ the initial and final plant height at $t = 0$ and $t = t_e$, respectively, $t_e$ the plant age at the end of growth and $t_m$ the plant age at the moment of maximal growth. Before curve fitting, the time points in day of year (DOY) at which the plant heights were measured were translated to plant ages $t$ in growing degree days (GDD), i.e. $t(i) = \sum_{j=0}^{j=i} max(T_j - T_b, 0)$ with $i$ the time point in DOY, $T_j$ the average air temperature at $j$ DOY (S1 Data) and $T_b = 5°C$ a base temperature below which no growth is assumed to occur [70, 85]. Optimization of the parameters $h_0$, $h_{max}$, $t_e$ and $t_m$ was done with the *nls* function in R using the 'port' algorithm. The maximum shoot growth rate was obtained by calculating the derivative of $h(t)$ (Eq 1) at $t_m$. After curve fitting, the values obtained for $t_m$ and $t_e$ were converted back from GDD to DOY and subsequently to DAS.

## Image-based phenotyping

Leaf 8 areas (81 DAS) were estimated by segmenting the flattened leaf images taken at the time of leaf harvest. The millimeter grid scale on each image was used to correct for perspective distortion and to create a uniform spatial resolution across the entire image of 100 pixels per cm. Images were cropped to remove the grid scale and sample label. Segmentation was done by training a U-Net convolutional neural network [86] on a small dataset of 25 images for which random patches of foreground and background were annotated using VGG Image Annotator (via) v:2.0.7 [87]. Random cropping, resizing, rotating (by multiples of 90 degrees), mirroring, color-jittering and gaussian blurring were applied to artificially increase the training dataset size. The training was done using the Adam optimizer [88] in Pytorch v:1.7.1 [89] with default settings. The pixel-wise cross-entropy loss was back-propagated only for annotated regions of each image. The learning rate was initially set to 1e-3 and was automatically halved as soon as the minimal training loss stagnated for more than 3 epochs. The network was trained for 16 epochs. The trained network was validated by visually evaluating it on unseen images, and then applied to all flattened leaf images.

Leaf 8 length and width at 81 DAS were measured on the flattened leaf images using ImageJ v:1.50 [90]. For measuring leaf 8 length, the midvein was traced from the leaf tip to the cutting point (i.e. where the petiole first lacked conspicuous laminar tissue) using the ImageJ segmented line tool. Leaf 8 width was measured at the widest point.

For measuring the projected area of the rosettes photographed at 42 DAS (i.e. the rosette imaging date closest to leaf sampling), a dedicated script was developed using the image analysis software HALCON (version 13.0.1.1, MVTec Software GmbH, Germany). First, the images were cropped to remove parts of adjacent plants visible on the pictures. To remove noise, both a gentle Gaussian filter and a median filter were applied. Each RGB image was then converted to the HSV color space, where the Hue channel was used to select the green plant parts using a threshold range for the green pixels (34–80) defined based on trial and error. Care was taken to also include the petioles. After this, a 'closing_circle' operator was used and remaining small lesions (due to insect damage) were filled up using the 'fill_up' operator. Only the largest segmented area was taken into account, to differentiate between the plant of interest and small weeds nearby.

The HALCON segmentation strategy worked well for the rosette images taken at 42 DAS, but regularly produced segmentation errors for images of smaller rosettes taken closer to the sowing date. An alternative segmentation approach was therefore used on rosette images taken at 14, 18, 21, 25, 28, 32, 35 and 39 DAS (and 42 DAS as control). The main difficulty for the earlier time points is distinguishing small rosettes from weeds and other distracting objects occurring on the field. This requires an algorithm with a larger field of view than what a HALCON script or standard U-net (see above) can provide. Instead, a standard pre-trained DenseNet M161 [91] was taken and augmented with additional bilinear upsampling layers after each 'dense' layer of the original algorithm. That is, the last feature layer of DenseNet was upsampled with bilinear interpolation and a weighted sum was made with the higher resolution 'dense' features. This was repeated for each dense layer up to the original input resolution. The network was trained for 175 epochs (final mean epoch loss = 0.01) on 54 hand-labeled images (6 images per time point) using stochastic gradient descent (SGD) with momentum (learning rate = 0.001 and momentum = 0.99). The learning rate was divided by 10 each time the train loss plateaued for more than 4 epochs. Image rotations, mirroring and HSV augmentations were used to augment the training data. The trained model was used to segment all rosette images. After segmentation, a post-processing step was performed to remove segmented parts of *B. napus* plants adjacent to the plant of interest and remaining weeds, using scikit-image v: 0.19.2 [92]. Only the connected component closest to the centroid of the image and other components within a 25-pixel distance of this central component (e.g. leaves of which the stalk was segmented incorrectly because of a lower chlorophyll content) were associated with the plant of interest. Connected components with an area less than 10,000 pixels were filtered out to eliminate small weeds. This approach was evaluated visually for all segmentations and proved to work well for most plants. Segmentations with missing plant parts or weeds that weren't filtered out by this post-processing step were manually corrected. A grey calibration card (Novoflex grey card 15 x 20 cm, NOVOFLEX Präzisionstechnik GmbH, Germany) was used as a reference to convert pixels to areas in $cm^2$. The projected rosette areas at 42 DAS estimated by this segmentation approach exhibit a Pearson correlation of 0.997 with the areas estimated by the aforementioned HALCON script.

## RNA sequencing

The frozen leaf samples for the 62 harvested non-border plants were grinded, and total RNA was extracted using the guanidinium thiocyanate-phenol-chloroform extraction method using TRI-reagent (Thermo Fisher Scientific) followed by DNA digestion using the RQ1 RNase-free DNase kit (Promega). ds cDNA was prepared using the Maxima H Minus Double-Stranded cDNA Synthesis Kit (#K2561, Thermo Fisher Scientific) to a concentration of ~17–38 ng/ul in 10mM Tris-Cl buffer (pH 8.5) at a minimum volume of 30ul. (~0.6–1.1 ug total). ds cDNA samples were sent to the University of Missouri Genomics Technology Core, where library preparation was performed (average insert size of 500 bp) using the Illumina TruSeq DNA PCR-Free Library Prep Kit according to the protocol described in [93]. 250 bp paired-end sequencing was performed at the Tufts University Genomics Core on an Illumina HiSeq 2500 machine in Rapid Run mode. The samples were sequenced in 3 batches (**S1 Data**).

The raw RNA-seq data was processed using a custom Galaxy pipeline [94] implementing the following steps. First, the fastq files were quality-checked using FastQC (v:0.5.1) [95]. Next, Trimmomatic (v:0.32.1) [96] was used to remove adapters, read fragments with average quality below 20 and trimmed reads shorter than 125 base pairs. The trimmed and filtered reads were mapped to the *Brassica napus* Darmor-bzh reference genome v:5 (https://www.genoscope.cns. fr/brassicanapus/data/) [83] using HISAT2 v:2.0.5 [97] with default values for all parameters.

Only the uniquely mapping reads or (in the case of multiple mappings) the best secondary alignment were kept for the following analyses. The mapping files were quantified using HTSeq v:0.6.1p1 [98] with the option 'Intersection-union', using the genome annotation of the *Brassica napus* Darmor-bzh reference genome v:5 (https://www.genoscope.cns.fr/brassicanapus/data/). No filtering steps were performed during preprocessing except for removing genes that were not expressed in any samples. Counts were normalized across samples and batches using a modified regularized log (*rlog*) model of the DESeq2 [99] package in R. Counts are still modeled in the same way as in the original *rlog* implementation, that is:

$$k_{ij} \sim NB(\mu_{ij}, \alpha_i)$$

$$\mu_{ij} = s_j \times q_{ij} \quad (Eq2)$$

$$\log_2(q_{ij}) = \mathbf{x}_j \cdot \boldsymbol{\beta}_i$$

Where $k_{ij} \in \mathbb{N}^+$ is the count of gene $i$ in sample $j$, which is assumed to be sampled from a negative binomial distribution (*NB*) with estimated mean $\mu_{ij} \in \mathbb{R}^+$ and estimated dispersion of the $i$th gene $\alpha_i$. $\mu_{ij}$ is taken as the expected count $q_{ij}$ for a 'typical' library size (i.e. with a size factor $s_j = 1$), scaled by a library size normalization factor $s_j$ for sample $j$. Note that $q_{ij}$ still contains batch effects: $\mathbf{x}_j \in \mathbb{R}^P$ is a vector of $p = 65$ predictors for sample $j$, including an intercept, 2 dummy variables for the smallest sequencing batches (1 and 3) that capture batch effects relative to the largest sequencing batch (2, the effects of which are absorbed in the intercept) and dummy variables for each of the 62 plants that were sampled. $\boldsymbol{\beta}_i \in \mathbb{R}^P$ contains the estimated coefficients for those predictors for gene $i$. As in [99], an empirical Bayes shrinkage procedure is used to estimate $\boldsymbol{\beta}_i$, using a flat prior for the intercept $\beta_{i0}$ and the sequencing batch coefficients, and a zero-centered normal prior for each plant coefficient $\beta_{ip_j}$ (with $p_j$ the index of the plant corresponding to sample $j$), with prior variance estimated using quantile matching as described in Love *et al.* [99]. There are only two differences compared to Love *et al.* [99]: the first is the addition of two batch coefficients as fixed effects in the design matrix, and the second is that log-fold changes used in the prior random effect variance computation are estimated relative to the mean of each batch instead of to the mean of all samples. Once the model is estimated, *rlog* counts are computed as in Love *et al.* [99], that is:

$$rlog_{ij} \equiv \beta_{i0} + \beta_{ip_j} \quad (Eq3)$$

Note that all samples $j$ belonging to the same plant (technical repeats) have the same value for $\beta_{ip_j}$. The modified *rlog* transformation removes library size effects and batch effects, unites technical repeats into one estimate and log-transforms the data (reducing heteroscedasticity) in a single step. In addition, using random effects for each plant allows pooling information from technical repeats while simultaneously basing variance estimates on all samples (including samples without technical repeats). This method therefore makes maximal use of the available data. The resulting data is show in **S11 Fig**.

## SNP detection and population structure analysis

Trimmed and filtered RNA-seq reads were aligned to the *Brassica napus* Darmor-bzh reference genome v:5 (https://www.genoscope.cns.fr/brassicanapus/data/) [83] using HISAT2 v:2.0.5 [97] with default values for all parameters. Genomic variants were detected for each plant using NGSEP v:3.3.2 [100] on the aligned reads. For downstream analyses, we focused on biallelic SNPs with a minimum genotype quality of 40 and called in at least 49 samples

(80% of the population). Missing calls were imputed using Beagle v:5.1 [101] using default parameters, and only SNPs with minor allele frequency (MAF) $\geq 0.05$ after imputation were kept, resulting in a dataset of 23,188 SNPs.

A neighbor-joining tree was made based on the SNP dataset with TASSEL v:5.2.60 [102], using 1-IBS (identity by state) as the distance measure while setting the distance from an individual to itself to zero. The tree was rendered using the polar tree layout in FigTree v:1.4.3 [103].

## Spatial autocorrelation analysis

Moran's I was calculated for each gene (phenotype) as $I = \frac{n}{w} \frac{(\mathbf{x}-\bar{\mathbf{x}})^T \mathbf{C}(\mathbf{x}-\bar{\mathbf{x}})}{\|(\mathbf{x}-\bar{\mathbf{x}})\|^2}$. Where $\mathbf{x}$ is a column vector of *rlog* gene expression (phenotype) values, $n$ is the number of samples and $w$ is the sum of elements of the connectivity matrix $\mathbf{C}$. For $\mathbf{C}$ a binary $n \times n$ queen contiguity-based spatial weight matrix was chosen, meaning that neighboring horizontal, vertical and diagonal plants are seen as connected. Note that $\mathbf{C}$ can differ from one phenotype to the next since not all phenotypes were available for all samples. For each gene (phenotype), the Moran's I was recalculated on $10^5$ random permutations of $\mathbf{x}$ to obtain an empirical null distribution, which was then compared to the real Moran's I to obtain a *p*-value. Finally, *p*-values were adjusted for multiple testing across all genes (phenotypes) using the BH procedure [104]. All calculations were done using the PySAL python library [105].

To assess the cause of the qualitative difference between the results obtained with the method above on the present dataset and the results obtained with a different method on a maize single-plant dataset in Cruz *et al.* [28], we also calculated Moran's I values and their significance as in Cruz *et al.* [28], using an inverse distance-based spatial weight matrix and parametric testing as implemented in the R package ape v:5.7 [106]. *p*-values were again adjusted for multiple testing across all genes (phenotypes) using the BH procedure [104].

## Variance analysis

Principal component analysis was done on various normalized versions of the gene expression count matrix and on the phenotype dataset (including qualitative phenotypes such as leaf 6 lesion severity (74 DAS) but excluding the plant height and rosette area time series except for the final time points, i.e. plant height (278 DAS) and rosette area (42 DAS)), using the 'prcomp' function in the R stats package on the centered gene expression datasets and the 'ppca' method in pcaMethods v:1.88.0 [107] on the z-scored phenotype dataset. Phenotype distributions were plotted using the 'histogram' function in Matlab R2018b with probability normalization option. Shapiro-Wilk and Anderson-Darling tests were performed using the 'normalitytest' script [108] and 'adtest' functions in Matlab R2018b, respectively. Outliers were defined as values more than three scaled median absolute deviations (MAD) away from the median, as is default in the Matlab R2018b 'isOutlier' function. Outliers were only removed for the purpose of calculating their effect on the phenotypes' normality and coefficient of variation, all other analyses used the complete phenotype dataset.

Normalized coefficients of variation (*normCVs*) for gene expression profiles were computed on batch and library size corrected data (without *rlog* transform). Normalized counts were obtained as $x_{ij} = k_{ij}/(\beta_{ib_j} \times s_j)$ where $\beta_{ib_j}$ is the batch effect for gene $i$ in sample $j$ as estimated in the *rlog* calculation (see above). Since batch 2 is absorbed in the intercept, $\beta_{ib_j} = 1$ for samples of batch 2. Contrary to the *rlog* transform, this method does not collapse technical repeats, and they were instead collapsed by averaging (as in **S11 Fig**, panel B, but without the $\log_2$-transform). From here on, variance analysis followed the same procedure as described in

Cruz *et al.* [28]. Briefly, a trendline was fitted to the $CV^2$ versus mean expression relationship (omitting genes expressed in < 10 samples) using a generalized linear model of the gamma family with identity link of the form $CV^2(\mathbf{x}) = {}^a/_{\bar{x}} + b$, with fitting parameters $a$ and $b$ [109] (**S12 Fig**). Code from the M3Drop R package [110] was used for this purpose. A normalized CV accounting for the observed mean-variance relationship was then calculated as $normCV(\mathbf{x}) = \log_2(CV^2(\mathbf{x})/trend(\bar{\mathbf{x}}))$ where $trend(\bar{\mathbf{x}})$ is the fitted value at the mean of $\mathbf{x}$.

## GO enrichment analysis

A Gene Ontology annotation for *Brassica napus* was generated using the TRAPID v.2.0 platform [111] with default parameters on April 16, 2020. Transcript sequences parsed from the *B. napus* Darmor-bzh reference genome annotation v:5 [83] using the gffread v.0.9.6 utility [112] were used as input for TRAPID, and PLAZA 4.5 dicots [113] was used as the reference database. GO enrichment *p*-values were calculated with hypergeometric tests and adjusted for multiple testing (*q*-values) using the BH procedure [104], either using custom R scripts or using BiNGO v:3.0.3 [114]. GO categories gravitating toward the top or bottom of gene lists ranked in order of decreasing Moran's I or normalized CV were detected using two-sided Mann-Whitney U tests (with genes belonging to the GO category of interest classified as group 1 and other genes as group 2), as implemented in the 'wilcox.test' function in the R stats package v:4.0.5, followed by BH *p*-value adjustment.

## Ortholog inference

Putative *A. thaliana* orthologs of *B. napus* genes were identified in two steps. First, putative orthologs of *B. napus* genes were identified in *B. rapa* and *B. oleracea* (source of the A and C subgenomes of *B. napus*, respectively), based on best similarity hits returned by TRAPID v.2.0 [111] and on the syntenic relationships reported in Chalhoub *et al.* [83] and Sun *et al.* [115]. Second, putative *A. thaliana* orthologs of the identified *B. rapa* and *B. oleracea* genes were retrieved from PLAZA 4.5 dicots [113], which provides orthology inferences integrating four different lines of evidence: orthogroup inference within gene families using OrthoFinder [116], orthology inference using gene tree-species tree reconciliation, orthology inference from best DIAMOND [117] hits and their inparalogs, and positional orthology inference through collinearity analysis [118]. The most likely *A. thaliana* orthologs of a given *B. napus* gene were taken to be the putative orthologs that are most strongly supported across both inference steps.

## Single-feature phenotype prediction models

**Single-gene models.** Linear mixed-effects models [119] were used to test gene expression-phenotype associations because they offer a robust statistical framework for significance testing on small sample sizes, even in the presence of potential spatial autocorrelation patterns in the data. Given a phenotype vector $\mathbf{y}$ and a vector $\mathbf{x}$ of a given gene's z-scored expression values across the field, we fit the following model:

$$\mathbf{y} = \beta_0 + \beta_1\mathbf{x} + \boldsymbol{\varepsilon}$$

$$\boldsymbol{\varepsilon} \sim \mathcal{N}(0, \Sigma) \tag{Eq4}$$

where $\beta_0$ is the intercept (expected phenotype value if the gene is not expressed), $\beta_1$ the gene effect coefficient, and $\boldsymbol{\varepsilon}$ the residual error which is assumed to follow a multivariate normal

distribution with a Gaussian covariance structure $\Sigma$ given by:

$$\Sigma_{ij} = \sigma_\varepsilon^2 \times \left( v \times I_{ij} + (1 - v) \times exp\left[ -\left( {d_{ij}}/{}_r \right)^2 \right] \right) \qquad \text{(Eq5)}$$

where $d_{ij}$ is the physical distance between plant $i$ and $j$ on the field, $\sigma_\varepsilon^2$ is the overall residual phenotype variance, the nugget $v$ (between 0 and 1) determines the proportion of the residual variance that is independently and identically distributed (*iid*) as opposed to governed by spatial autocorrelation, the range $r$ determines how fast the residual phenotype correlation between plants drops when the distance between them increases, and $I$ is an identity matrix. The same model form was used to predict final yield phenotypes, e.g. total seed weight, as a function of one of the phenotypes measured early in the growing season, e.g. leaf 8 area (81 DAS). All parameters ($\beta_0, \beta_1, \sigma_\varepsilon, v, r$) are estimated from the data by Restricted Maximum Likelihood (ReML) estimation, implemented in the nlme package [120] in R. In some cases the lme model didn't converge and a regular linear model (lm) was used instead. *p*-values for the $\beta_1$ coefficients were determined using Wald tests and adjusted for multiple testing using the BH procedure [104].

For each of the 100 genes with the lowest $\beta_1$ *q*-value for a given phenotype, a 9-times repeated 10-fold cross-validation scheme was used to assess the gene's predictive power (see section on multi-gene models for details). The median test $R^2$ score across all 90 splits was used as a measure of prediction performance. This enables fair comparison between the prediction performance of single-gene and multi-gene models.

**Single-phenotype models.**   The same linear mixed-effects modeling and cross-validation strategy as used for the single-gene models was also used also to model spring phenotypes as a function of autumnal leaf or rosette phenotypes. Leaf 6 and leaf 8 phenotypes and the rosette area at 42 DAS were used as features for predicting all spring phenotypes. In a separate analysis, also earlier rosette areas (14–42 DAS) were used as features, in order to assess how the predictive power of the projected rosette area for yield phenotypes evolves over time.

**Alternative single-gene models for ratio phenotypes.**   For seeds per silique (on stem 1 or the entire plant), the following alternative log-link model was fitted using the nlme package [120] in R:

$$\ln(E(\mathbf{n} \oslash \mathbf{d})) = \beta_0 + \beta_1 \mathbf{x} \qquad \text{(Eq6)}$$

where $\oslash$ stands for the element-wise division of the numerator $\mathbf{n}$, a vector containing the seed count stem 1 for all plants, by the denominator $\mathbf{d}$, a vector containing the silique count stem 1 for all plants. $\mathbf{x}$ is the expression profile of a given gene across plants. The numerator is assumed to follow a normal distribution given the denominator $\mathbf{d}$ and the gene expression profile:

$$\mathbf{n} \sim \mathcal{N}(\mathbf{d} \cdot \exp(\beta_0 + \beta_1 \mathbf{x}), \Sigma) \qquad \text{(Eq7)}$$

Various error models $\Sigma$ were tried out. For each gene, $\Sigma$ is either a constant $\sigma^2$ across all plants estimated from the data, a spatially covarying error structure (using a Gaussian covariance structure as for the other single-gene models, see above), a heteroscedastic error structure with the error variance increasing linearly with the estimate, or a both spatially covarying and heteroscedastic error structure. The parameters $\beta_0, \beta_1, \sigma^2$ (and optionally the nugget and range for spatial models) were estimated using the 'ngls' function in nlme. *p*-values for the gene expression coefficients $\beta_1$ were determined using Wald tests and adjusted for multiple testing using the BH procedure [104].

A similar model was used for the branches per stem phenotype:

$$\ln(E((c + \mathbf{n}) \oslash \mathbf{d})) = \beta_0 + \beta_1 \mathbf{x} \tag{Eq8}$$

where $\mathbf{n}$ is a vector containing the total branch count for all plants, $\mathbf{d}$ is a vector containing the stem count for all plants, and c is an extra offset introduced to account for the amount of branches per stem decreasing with increasing numbers of stems on a plant.

## Multi-feature phenotype prediction models

**Multi-gene models.** Predictive models were made for each phenotype based on z-scored *rlog* gene expression data, using either all genes or only transcription factors as potential features. Random forest [34] and elastic net [33] models were constructed with scikit-learn v:0.23.2 [121] using a 10-fold cross-validation scheme. Model learning on the training data in each cross-validation split was done in two steps. First a feature selection model was used to select promising features, and then a RF or enet model was built on the selected features. Three methods were used as alternatives for feature selection. The first feature selection technique used was HSIC lasso [35] as implemented in the pyHSICLasso package [122], which generally selected at most 200 genes. The second feature selection technique was a filter selecting gene expression profiles exhibiting a significant Spearman correlation with the phenotype of interest ($q \leq 0.01$; if no features survived this filter, the threshold was set at $p \leq 0.001$). The third feature selection technique was a filter selecting genes with *rlog* gene expression $> 0$ in at least half of the samples (median *rlog* gene expression $> 0$). enet models were built using a fourfold inner cross-validation loop to estimate the model hyperparameters. For RF models, 1000 trees were estimated (n_estimators = 1000) using bootstrapping (bootstrap = True), and $\sqrt{n}$ features (with $n$ the total number of features) were considered when looking for the best split (max_features = "auto"). The hyperparameters 'max_depth' (the maximum number of nodes) and 'min_samples_leaf' (the minimal number of samples at each leaf node) were optimized using a grid search with possible values (1, 2, 5, 10, 20, 50) and (1, 2, 5) for 'max_depth' and 'min_samples_leaf', respectively. Optimal hyperparameters were selected based on generalization scores on out-of-bag (oob) samples (oob_score = True).

For each combination of phenotype, machine learning method and feature selection technique, 9 repeats of the aforementioned 10-fold cross-validation scheme were performed, giving rise to 90 train-test data splits in total. For each split, an out-of-sample (oos) $R^2$ score was computed from the predicted and observed phenotype values in the test set, and the median oos $R^2$ across all 90 splits (= median test $R^2$) is reported as a measure of model prediction performance. Alternative $R^2$ values and PCC values were computed based on the combined set of test predictions across all 10 splits of a cross-validation repeat. The medians of those $R^2$ and PCC values across the 9 cross-validation repeats for a given model are reported as the median pooled $R^2$ and median pooled PCC score of the model, respectively.

For both enet and RF models, genes of potential interest for a given phenotype were ranked based on their median importance across the 90 cross-validation splits of the model version with the highest median test $R^2$ score (the difference between model versions being the use of different feature selection techniques). For RF models, the gini importance of a gene was used as its importance score. For enet models, the absolute value of a gene's estimated model coefficient was used.

**Models on permuted datasets.** For all continuous and high-count phenotypes and for both the 'all genes' and 'transcription factors' feature sets, models were trained and tested on 90 datasets in which the phenotype values were permuted, using the same machine learning method and feature selection technique as for the model with the best median test $R^2$ score on

real data for the given phenotype and feature set. For each phenotype and feature set, one model was trained per permuted dataset, using a single 90–10 train-test split mimicking one fold of the cross validation setup used on real data.

**Multi-phenotype models.** For all phenotypes measured in spring, additional predictive models were made based on z-scored data for 14 leaf and rosette phenotypes measured in the preceding autumn. We used the same modeling approach as for the expression-based models (RF and enet, 9 repeats of 10-fold nested cross-validation), except that the feature selection step of the expression-based modeling protocol was skipped given the low number of potential model features. In this respect, using enet models instead of a simple linear regression framework is technically also unnecessary, but enets were used nevertheless to maximize comparability of the early phenotype-based and expression-based modeling results.

## Supporting information

**S1 Fig. SNP analysis of individual plants.**
(PDF)

**S2 Fig. Correlations of pairwise distances between plants across different omics layers.**
(PDF)

**S3 Fig. Phenotype field plots.**
(PDF)

**S4 Fig. Phenotype histograms.**
(PDF)

**S5 Fig. Heatmap of Moran's I versus normCV values of gene expression profiles.**
(PDF)

**S6 Fig. Phenotype predictions versus observations.**
(PDF)

**S7 Fig. Performance of log-link model predicting seed count stem 1 conditioned on silique count stem 1 as a function of expression of the best predictor gene, *BnaCnng56980D*.**
(PDF)

**S8 Fig. Performance of log-link model predicting total branch count conditioned on stem count as a function of expression of the best predictor gene, *BnaC01g26820D*.**
(PDF)

**S9 Fig. Top predictor genes in enet models of leaf phenotypes.**
(PDF)

**S10 Fig. Top predictor genes in RF models of yield phenotypes.**
(PDF)

**S11 Fig. Sequencing batch effects on RNA-seq count data.**
(PDF)

**S12 Fig. Gene expression variability in the *B. napus* single-plant dataset.**
(PDF)

**S1 Data. Plant metadata, weather station data, normalized gene expression profiles and phenotype profiles.**
(XLSX)

**S2 Data. Moran's I values for gene expression and phenotype profiles and associated GO analysis results.**
(XLSX)

**S3 Data. Coefficients of variation for gene expression and phenotype profiles and associated GO analysis results.**
(XLSX)

**S4 Data. Transcripts significantly associated with phenotypes.**
(XLSX)

**S5 Data. GO enrichment of gene and TF sets significantly associated with phenotypes.**
(XLSX)

**S6 Data. Prediction performance scores and feature importance scores for single-gene and multi-gene models.**
(XLSX)

**S7 Data. Literature support for top-10 TF predictors of leaf and yield phenotypes.**
(XLSX)

**S8 Data. Prediction performance scores and feature importance scores for single-phenotype and multi-phenotype models.**
(XLSX)

**S9 Data. Prediction performance scores for models predicting mature phenotypes from early rosette areas (14 DAS—42 DAS).**
(XLSX)

## Acknowledgments

The authors thank Benjamin Wittkop and Rod Snowdon for providing *B. napus* Darmor seeds, Luc van Gyseghem, Thomas Vanderstocken and Katleen Sucaet for their help with setting up and maintenance of the field trial, Dorota Herman and Kirin Demuynck for assistance with leaf sampling, and Chris Pires for advice on sample prep for RNA sequencing.

## Author Contributions

**Conceptualization:** Steven Maere.

**Data curation:** Sam De Meyer, Daniel Felipe Cruz, Peter Lootens, Steven Maere.

**Formal analysis:** Sam De Meyer, Daniel Felipe Cruz, Tom De Swaef, Peter Lootens, Michael Van de Voorde, Stijn Hawinkel, Steven Maere.

**Funding acquisition:** Steven Maere.

**Investigation:** Sam De Meyer, Daniel Felipe Cruz, Tom De Swaef, Peter Lootens, Jolien De Block, Kevin Bird, Heike Sprenger, Michael Van de Voorde, Stijn Hawinkel, Tom Van Hautegem, Hilde Nelissen, Isabel Roldán-Ruiz, Steven Maere.

**Methodology:** Sam De Meyer, Tom De Swaef, Peter Lootens, Isabel Roldán-Ruiz, Steven Maere.

**Project administration:** Steven Maere.

**Resources:** Dirk Inzé, Hilde Nelissen, Isabel Roldán-Ruiz.

**Software:** Sam De Meyer, Peter Lootens, Steven Maere.

**Supervision:** Dirk Inzé, Hilde Nelissen, Isabel Roldán-Ruiz, Steven Maere.

**Validation:** Steven Maere.

**Visualization:** Sam De Meyer, Daniel Felipe Cruz, Steven Maere.

**Writing – original draft:** Sam De Meyer, Daniel Felipe Cruz, Steven Maere.

**Writing – review & editing:** Tom De Swaef, Peter Lootens, Jolien De Block, Kevin Bird, Heike Sprenger, Michael Van de Voorde, Stijn Hawinkel, Tom Van Hautegem, Dirk Inzé, Hilde Nelissen, Isabel Roldán-Ruiz, Steven Maere.

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
