## [Decision Letter · Decision Letter 0]

9 Feb 2023

Dear Prof. Maere,

Thank you very much for submitting your manuscript "Predicting yield of individual field-grown rapeseed plants from rosette-stage leaf gene expression" for consideration at PLOS Computational Biology. As with all papers reviewed by the journal, your manuscript was reviewed by members of the editorial board and by several independent reviewers. The reviewers appreciated the attention to an important topic. Based on the reviews, we are likely to accept this manuscript for publication, providing that you modify the manuscript according to the review recommendations.

Sincerely,

Kiran Raosaheb Patil, Ph.D.

Section Editor

PLOS Computational Biology

Reviewer's Responses to Questions

**Comments to the Authors:**

Reviewer #1: In this interesting paper, the authors use single plant-omics to predict yield of individual field-grown rapeseed plants from rosette-stage 2 leaf gene expression. This is interesting as it suggests a way of predicting crop yield from the expression state of younger plants. They use machine learning to successfully predict the phenotypes of individual B. napus plants from rosette-stage leaf gene expression data. The authors have generated a huge amount of data on individual plants and have carefully compared early stage gene expression and phenotype with final yield. It is nice that the top predictive genes from their models are often linked to the juvenile-adult phase change in Arabidopsis. I think the topic and material will be of interest to PloS computational Biology readers, and the data will be useful to researchers who want to dig into the topic. The authors should be commended for making so much of their data and code available online. I have the following points:

Major point.

My one major problem with the paper is I found it difficult to understand from the main text how the machine learning approach was applied, how the techniques were chosen, and to get an intuitive understanding of the results and how they can be validated. This wasn’t completely fixed by going to the methods, which explained in detail the approach, but not how the approach was chosen in the first place. This can be simply fixed with a few more explanations.

For example, the authors write: ‘We built random forest (RF) and elastic net (enet) models to predict the phenotypes of individual plants from their autumnal leaf 8 transcriptome, using either all genes or only transcription factors (TFs) as potential features and using three different feature selection techniques (see Methods). For each combination of phenotype, model type (RF or enet), potential feature set (all genes or TFs) and feature selection technique, 9 repeat models werelearned, each time using 10-fold cross-validation with different splits (see Methods), resulting per combination in a total of 90 test sets and 9 test set predictions per plant.

It should be explained in the main text the differences between the single gene and multi-gene models, and why random forest and elastic net models are used (why both?). It would also be helpful to explain how the multi-gene models are implemented and what assumptions were used.

Minor points

The authors write in lines 193-206, ‘ In a previous study on a similar number of field-grown maize plants [28], 14.17% of transcripts were found to be significantly spatially autocorrelated at q ≤ 0.01, which is considerably more than the 0.22% recovered here at q ≤ 0.05. This may be due to differences in the way Moran’s I values and their significance were calculated in Cruz, De Meyer (28) versus the present study (see Methods).

Reference 28 appears to be the from the same group as this paper. Wouldn’t it be possible to get the data from 28 and do the same test? The methods dont seem to explain the differences in how the Moran's I values were calculated between the two papers I don’t fully understand what was done after this in lines up to 206, and what the evidence is that there is spatial patterning in the new data? I don’t think it matters for this papers results whether there is spatial patterning, but I found this part of the paper confusing.

Lines 935/936 I found it interesting that seedling emergence was a poor predictor of yield ‘ the seedling emergence date was not recorded in the present field trial, but the closest proxy that was measured, namely rosette area at 14 DAS, was found to be a bad predictor for yield, indicating that variation in seed germination and seedling emergence across the field did not by themselves have a major impact on yield.’ I note in the methods that ‘Early- and late-emerging seedlings were pruned preferentially (based on visual assessment) to make the remaining seedling population as homogeneous as possible.’ Should the authors clarify that in this work they cant assess the effects of seedling emergence on yield as they prune early and late emerging seedlings? Also, is the data showing that rosette area at 14DAS was a bad predictor for yield in the paper? (sorry if I missed it, but if so maybe reference where this data is shown here as well?)

Reviewer #2: The study by De Meyer et al., investigates the phenotypes and gene expression in rapeseed plants, with the aim to predict spring phenotypes given autumn data. The authors found that single-plant omics can be used to identify genes and processes influencing crop yield in the field. Overall, the work is interesting, well-executed, and the topic exciting. Overall, we would be happy to see this paper published.

However, there are some places that need clarification:

Figure 1: ‘Principal component analysis (PCA) suggests that there are no subpopulations of plants with distinct expression or phenotype profiles’. It would be good to have statistics support this. For example, is there a correlation between the distance of plants in the field and the points in the PCA plot?

Line 195: The authors mention that Moran’s I is calculated differently in this study. Why? Please elaborate.

Line 452 (and other occurrences): The autors discuss the differences in performance of single- and multi-gene models, but it is unclear what we should look for. Please cite the table/figure, and also the R2 value ranges.

The top predictors for leaf and seed phenotypes chapter is somewhat long and tedious to read due to the descriptions of many genes found in the table. Perhaps it would be easier to summarize the findings and phenotypes as another table/figure?

Line 429: Is a comparison of the phenotype prediction performance between multi-gene models and transcription factors models relevant?

Line 455-456: According to the data in Table 2, it seems that most of the shoot dry weight traits are not better predicted by muti-gene models than single-gene models.

It would be useful to indicate TFs that are found in RF and enet models if Figure 3, 4.

Line 495: “Random forest” and “elastic net” abbreviations have been defined in line 330, so no need to define it again in 495. Please also check similar problems throughout the manuscript.

Line 695: Please mark the numbers of TFs in Figure 3 and Figure S8 as well.

Is it possible to further analyze whether top predictors and phenotype are positively or negatively correlated?

**Have the authors made all data and (if applicable) computational code underlying the findings in their manuscript fully available?**

Reviewer #1: Yes

Reviewer #2: Yes

PLOS authors have the option to publish the peer review history of their article (what does this mean?). If published, this will include your full peer review and any attached files.

Reviewer #1: No

Reviewer #2: **Yes: **Marek Mutwil

Figure Files:

Data Requirements:

Reproducibility:

References:

---

## [Decision Letter · Decision Letter 1]

5 May 2023

Dear Prof. Maere,

We are pleased to inform you that your manuscript 'Predicting yield of individual field-grown rapeseed plants from rosette-stage leaf gene expression' has been provisionally accepted for publication in PLOS Computational Biology.

Best regards,

Kiran Raosaheb Patil, Ph.D.

Section Editor

PLOS Computational Biology

Reviewer's Responses to Questions

**Comments to the Authors:**

Reviewer #1: I am happy with the revisions and support publication

Reviewer #2: The authors did a great job addressing our questions.

**Have the authors made all data and (if applicable) computational code underlying the findings in their manuscript fully available?**

Reviewer #1: None

Reviewer #2: None

PLOS authors have the option to publish the peer review history of their article (what does this mean?). If published, this will include your full peer review and any attached files.

Reviewer #1: No

Reviewer #2: **Yes: **Marek Mutwil

---

## [Editor Report · Acceptance letter]

23 May 2023

PCOMPBIOL-D-23-00014R1 

Predicting yield of individual field-grown rapeseed plants from rosette-stage leaf gene expression

Dear Dr Maere,

I am pleased to inform you that your manuscript has been formally accepted for publication in PLOS Computational Biology. Your manuscript is now with our production department and you will be notified of the publication date in due course.

With kind regards,

Timea Kemeri-Szekernyes
